

# SCDNA: a serially complete precipitation and temperature dataset for North America from 1979 to 2018

Guoqiang Tang[1,2], Martyn P. Clark[1,2], Andrew J. Newman[3], Andrew W. Wood[3], Simon Michael Papalexiou[2,4], Vincent Vionnet[5], and Paul H. Whitfield[1,2]

[1]University of Saskatchewan Coldwater Lab, Canmore, Alberta, Canada

[2]Centre for Hydrology, University of Saskatchewan, Saskatoon, Saskatchewan, Canada

[3]National Center for Atmospheric Research, Boulder, Colorado

[4]Department of Civil, Geological and Environmental Engineering, University of Saskatchewan, Saskatchewan, Canada

[5]Environmental Numerical Research Prediction, Environment and Climate Change Canada, Dorval, Quebec, Canada

*Correspondence to*: Guoqiang Tang (guoqiang.tang@usask.ca)

**Abstract:** Station-based serially complete datasets (SCDs) of precipitation and temperature observations are important for hydrometeorological studies. Motivated by the lack of serially-complete station observations for North America, this study seeks to develop a SCD from 1979 to 2018 from station data. The new SCD for North America (SCDNA) includes daily precipitation, minimum temperature ($T_{min}$), and maximum temperature ($T_{max}$) data for 27280 stations. Raw meteorological station data were obtained from the Global Historical Climate Network Daily (GHCN-D), the Global Surface Summary of the Day (GSOD), Environment and Climate Change Canada (ECCC), and a compiled station database in Mexico. Stations with at least 8-year records were selected, which underwent location correction and were subjected to strict quality control. Outputs from three reanalysis products (ERA5, JRA-55, and MERRA-2) provided auxiliary information to estimate station records and were also used as an assessment benchmark. Infilling during the observation period and reconstruction beyond the observation period were accomplished by combining estimates from 16 strategies (variants of quantile mapping, spatial interpolation, and machine learning). A sensitivity experiment was conducted by assuming 30% observations of stations were missing – this enabled independent validation and provided a reference for reconstruction. Quantile mapping and mean-value corrections were applied to the final estimates. The median Kling-Gupta efficiency (KGE) values of the final SCDNA for all stations are 0.90, 0.98, and 0.99 for precipitation, $T_{min}$ and $T_{max}$, respectively. The SCDNA is closer to station observations than four benchmark gridded product, and can be used in applications that require either quality-controlled meteorological station observations or reconstructed long-term estimates for analysis and modelling. The dataset is available at https://doi.org/10.5281/zenodo.3735534 (Tang et al., 2020).

**Key words:** serially complete dataset; precipitation; temperature; North America





## 1 Introduction

Station-based serially complete datasets (SCDs, see Table A1 for all acronyms) are important for meteorological, climatological and hydrological studies (Kanda et al., 2018; Ramos-Calzado et al., 2008), such as the production of retrospective gridded products (Di Luzio et al., 2008; Kenawy et al., 2013; Newman et al., 2019; Serrano-Notivoli et al., 2019), trend analysis (Knowles et al., 2006; Anderson et al., 2009; Papalexiou and Montanari, 2019), and climatologic index calculation (Alexander et al., 2006; Papalexiou et al., 2018). These SCDs are useful because station-based observational often contain missing values due to factors such as observer absence, instrumental failures and interrupted communication (Hasanpour Kashani and Dinpashoh, 2012). Moreover, station observations failing quality control tests such as outlier and homogeneity checks may not be reliable (Menne et al., 2012), and many stations are only maintained over a relatively short period of time or portions of the year, resulting in data gaps that could affect the analysis of climate variability or long-term trends (Rubin, 1976; Stooksbury et al., 1999). Serial completeness is also a critical requirement for real-time station-based applications, which regularly contend with missing data values due to latencies in station reporting, quality control and processing (Tang et al., 2009).

Many methods have been developed to estimate missing observations and reconstruct time series of stations; they can be grouped in self-contained infilling, spatial interpolation, quantile mapping (QM), and machine learning methods.

1. Self-contained infilling only uses records of the target station to estimate its own missing values. Typical methods include interpolation based on data from previous and subsequent days or replacing missing values by long-term mean (Kemp et al., 1983; Pappas et al., 2014). Self-contained infilling, however, only performs well for variables with high temporal autocorrelation such as temperature and is problematic for daily precipitation (Simolo et al., 2010; Teegavarapu and Chandramouli, 2005), and in covering lengthy gaps.

2. Spatial interpolation uses neighboring stations (identified on spatial distance or statistical similarity) to estimate data at the target station, which can be divided into two types: the first uses information only from neighboring stations; and common methods include linear interpolation and inverse distance weighting (IDW; Shepard, 1968). The second method needs information from both neighboring and target stations. Typical examples include the revised normal ratio (NR; Young, 1992) and the single best estimator (Eischeid et al., 1995, 2000), which use correlation coefficients (CCs) between target and neighboring stations to estimate merging weights. This second type of spatial interpolation also includes more sophisticated methods (e.g., multiple linear regression, optimal interpolation, and kriging) that build a functional relationship between neighboring and target stations (Simolo et al., 2010). Previous studies have shown that multiple linear regression based on the least absolute deviation criteria (MLAD) performs better than many interpolation methods such as IDW, NR, and optimal interpolation in infilling/reconstruction (Eischeid et al., 2000; Kanda et al., 2018).

3. QM is widely used to correct bias of meteorological data (Maraun, 2013; Cannon et al., 2015) and performs well in estimating missing station data (Simolo et al., 2010; Newman et al., 2015, 2019; Devi et al., 2019). In QM-based estimation, the cumulative distribution functions (CDFs) of observations from neighboring and target stations are





derived, and the record at the target station is estimated as the inverse of its CDF using concurrent CDF probability
information from neighboring stations. QM can avoid the problem of overestimating wet days in precipitation series
and preserve the frequency distribution of time series, which is useful for estimating extreme events (Cannon et al.,

69    2015).

4. Machine learning techniques have been successfully applied to infill station record gaps (Dastorani et al., 2010;
Wambua et al., 2016). For example, Coulibaly and Evora (2007) estimated missing daily precipitation and
temperature in northeastern Canada using six types of artificial neural networks (ANNs). Ustaoglu et al. (2008)
estimated daily temperature using three ANN methods in the Geyve and Sakarya basin, Turkey. Gene expression
programming was applied in the estimation of missing monthly rainfall data in Malaysia (Che Ghani et al., 2014).
Sattari et al. (2017) recommended that a decision-tree algorithm can be used to estimate monthly precipitation due
to its simplicity and high accuracy. Serrano-Notivoli et al. (2019) applied the $k$-nearest neighbours regression to
reconstruct minimum temperature ($T_{min}$) and maximum temperature ($T_{max}$) observations in Spain to form a gridded
dataset.
Previous SCDs have been developed using multiple infilling and reconstruction methods. For instance, Eischeid et al.
(2000) produced a daily SCD from 1951 to 1991 for the western United States (U.S.), including 2962 precipitation
stations and 2034 temperature stations; Vicente-Serrano et al. (2003) produced a daily SCD from 1901 to 2002 for
northeast Spain using 3106 precipitation stations; Di Piazza et al. (2011) built a monthly SCD from 1921 to 2004 for
Sicily, Italy using 247 precipitation stations; and Woldesenbet et al. (2017) produced a daily SCD of precipitation and
temperature from 1980 to 2013 for the Upper Blue Nile Basin using six stations. There is currently no SCD for North
America; this means that researchers often must collect station data from different databases, which is time-consuming
and may cause inconsistencies between studies based on different methods.
Responding to this need, we develop a retrospective 40-year daily SCD for North America (SCDNA) of precipitation,
$T_{min}$ and $T_{max}$ from 1979 to 2018. Central America and Caribbean are also covered by SCDNA. Station observations
are collected from four global and regional databases and undergo strict quality control to eliminate dubious records.
Since the performance of infilling and reconstruction methods differs in space and time, the results from 16 strategies
are merged to produce a single deterministic estimate. Finally, the SCDNA is compared to four gridded products to
demonstrate its performance and areas for improvement. The SCDNA is expected to have a wide variety of
applications in North America, and the methodology can be used to produce SCDs in other regions of the world.
**2 Datasets**
**2.1 Meteorological station data**
This study uses precipitation, $T_{min}$, and $T_{max}$ station data from four databases, the Global Historical Climate Network
Daily (GHCN-D; https://www.ncdc.noaa.gov/ghcnd-data-access; Menne et al., 2012), the Global Surface Summary
of the Day (GSOD; https://catalog.data.gov/dataset/global-surface-summary-of-the-day-gsod), Environment and



99 Climate Change Canada (ECCC; https://climate.weather.gc.ca/historical_data/search_historic_data_e.html), and the

100 Mexico database from Servicio Meteorológico Nacional, under the Comisión Nacional del Agua (Livneh et al., 2015).

101 Only stations with at least 8-year precipitation or $T_{min}$ and $T_{max}$ records between 1979 to 2018 are utilized. The

102 requirement for minimum recording length is different among studies (e.g., Eischeid et al., 2000; Newman et al., 2015).

103 We adopted a relatively short time limitation because (1) 8-year records are sufficient to provide basic support for

104 missing value estimation (Fig. S1), and (2) the open-access dataset and codes enable users to design customized data

105 selection criteria according to their research requirements.

106 The numbers of stations with at least 8-year records are 33026, 4619, 3634, and 4049 for GHCN-D, GSOD, ECCC,

107 and the Mexico database, respectively (Table 1). Their spatial distributions are shown in Fig. S2. GHCN-D has

108 complied a large amount of data from many sources including the Mexico database and ECCC. For identical stations

109 from different sources, we keep the one with longer observation history, resulting in the exclusion of ~95% of stations

110 from the Mexico database and adoption of ~91% of stations from ECCC. Stations with more than 30% missing values

111 in the observation period are excluded because they could be seasonal stations or suffer serious instrumentation

112 problems. Stations overlapping in space (same latitude and longitude) and without sufficient metadata for

113 discrimination are merged (see Sect. 3.2). The above screening reduces the available stations from 45328 to 31772

114 (Table 1), yet more stations are discarded due to quality control procedures (Sect. 3.1). The final SCDNA includes

115 24721 precipitation, 19677 $T_{min}$, and 19684 $T_{max}$ stations; note that the numbers of $T_{min}$ and $T_{max}$ stations differ as

116 quality controls can result in excluding the one and reserving the other in some stations.

117 Most stations are located in the Contiguous United States (CONUS), southern Canada, and Mexico, while few stations

118 are located in high-latitude regions such as the Arctic Archipelago (Fig. 1b and c). The spatial distributions of

119 precipitation and temperature stations are similar, except in eastern CONUS where precipitation stations have a higher

120 density.

121 Table 1. Numbers of stations with at least 8-year records from 1979 to 2018

| Station numbers | GHCN-D | GSOD | ECCC | Mexico | Merge | Total |
|---|---|---|---|---|---|---|
| Original numbers | 33026 | 4619 | 3634 | 4049 | 0 | 45328 |
| SCDNA input | 24765 | 4331 | 3100 | 187 | 207 | 31772 |
| SCDNA output: precipitation | 19255 | 2656 | 2440 | 170 | 200 | 24721 |
| SCDNA output: $T_{min}$ | 13445 | 3650 | 2219 | 167 | 196 | 19677 |
| SCDNA output: $T_{max}$ | 13453 | 3651 | 2217 | 167 | 196 | 19684 |

122 Notification: "Merge" is derived from stations with overlapped locations from all the other data sources (Sect. 3.1.1).

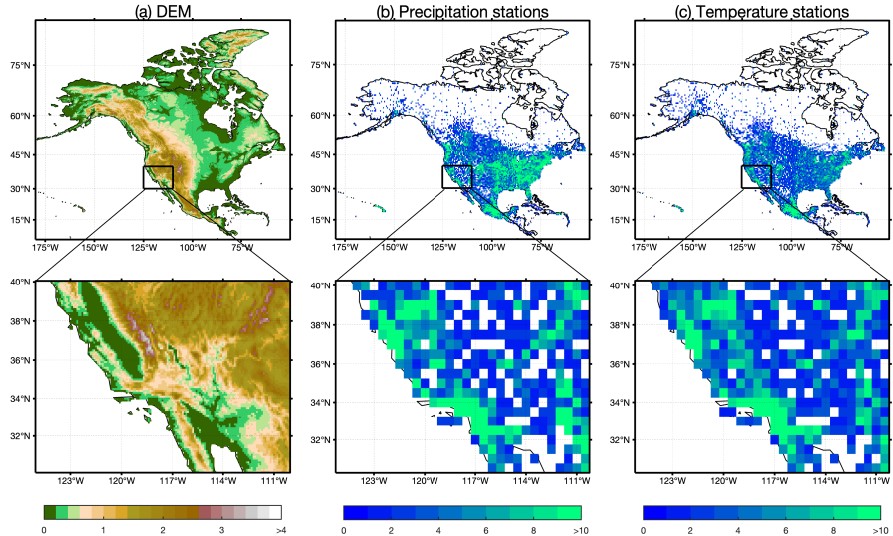

123

Figure 1. (a) Digital elevation model (DEM; Sect. 2.3) of North America. (b) and (c) are the densities of stations at
the 0.5°×0.5° resolution for precipitation and temperature, respectively. $T_{min}$ and $T_{max}$ stations are highly consistent,
and thus $T_{min}$ is used to represent temperature in (c). The nested black boxes show examples of DEM and station
densities.

In North America, more station observations occur in U.S. than in Canada and Mexico (Fig. 2). The number of samples
in U.S. increases from 1979 to 2018, and there are more precipitation samples than temperature samples. For Canada,
the numbers of precipitation and temperature samples are similar and show a decrease from 1988 to 2018; the sample
number in 2018 is only 61.76% of that in 1988. Mexico has more meteorological samples than Canada, yet this number
decreases after 1983. The decreasing trend is especially sharp after 2012 which may be due to the delay in data
collection or termination of some stations.

Figure 3 shows the fractions of missing values for all stations during the observation period (referred as ratio-1) and
during the entire period from 1979 to 2018 (referred as ratio-2). For temperature, ~20% of the stations have more than
20% missing values in the observation period (ratio-1), and ~20% of the stations have more than 70% missing values
in the entire period (ratio-2). For precipitation, the fraction of missing values is larger. The fractions show strong
spatial variations (Fig. S3). Ratio-2 is smaller for precipitation stations in western U.S. and temperature stations in
central U.S., but larger in Canada and Alaska. Most stations in Mexico have higher ratio-1 than other regions in North
America, indicating that those stations have notable fractions of missing values during the observation period.

In summary, the curves of ratio-1 indicate that a small number of missing values need infilling during the observation
period, while the curves of ratio-2 indicate that extensive reconstruction is needed over the entire period.

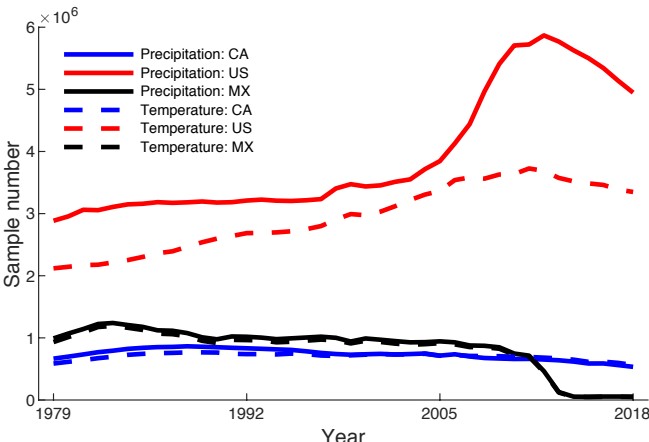

143

Figure 2. Sample numbers of stations for each year from 1979 to 2018. CA represents Canada, US represents United States, and MX represents Mexico. $T_{max}$ stations are highly consistent with $T_{min}$ stations, and thus $T_{min}$ is used to represent temperature. The numbers of samples could be a better indicator than the numbers of stations because many stations have notable missing values.

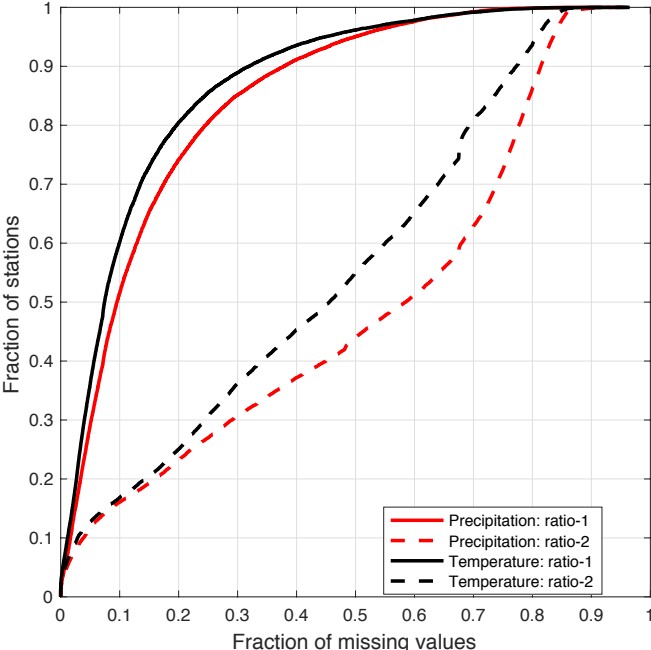


Figure 3. The fraction of missing values for stations with at least 8-year records. Ratio-1 is the degree of missingness during the observation period, and ratio-2 is the degree of missingness during the entire period of interest (1979 to 2018). $T_{min}$ is used to represent temperature because $T_{max}$ show almost overlapped curves with $T_{min}$.


## 2.2 Reanalysis products


We use reanalysis precipitation, $T_{min}$ and $T_{max}$ from the fifth generation of European Centre for Medium-Range
Weather Forecasts (ECMWF) atmospheric reanalyses of the global climate (ERA5; Copernicus Climate Change
Service (C3S), 2017), the Japanese 55-year Reanalysis (JRA-55; Kobayashi et al., 2015), and the Modern-Era
Retrospective analysis for Research and Applications, Version 2 (MERRA-2; Gelaro et al., 2017) (see Table 2). The
ERA5 and JRA-5 do not provide daily outputs, thus, daily precipitation is accumulated from sub-daily estimates while
daily $T_{min}$ and $T_{max}$ are estimated by the sub-daily minimum and maximum temperature values. Gridded reanalysis
precipitation is linearly interpolated to match point-scale station data, and $T_{min}$ and $T_{max}$ are downscaled using
temperature lapse rate (TLR; see Sect. 3.1).
Table 2. Information on the three reanalysis products.

| Products | Spatial resolution | Temporal resolution | Period | Agency |
|---|---|---|---|---|
| ERA5 | 0.25°×0.25° | 1 h | 1979-present | European Centre for Medium-Range Weather Forecasts |
| JRA-55 | ~60 km | 3 h | 1958-present | Japan Meteorological Agency |
| MERRA-2* | 0.5°×0.625° | daily | 1980-present | NASA's Global Modeling and Assimilation Office |

* MERRA-2 provides outputs in temporal resolutions from 1 h to 1 month; here we use daily values.

## 2.3 Auxiliary data


The Multi-Error-Removed Improved-Terrain digital elevation model (MERIT DEM) at a 3 sec (~90 m at the equator)
resolution (Yamazaki et al., 2017) is used in this study. To enable temperature downscaling, the high-resolution DEM
is spatially averaged to the original resolutions of ERA5, MERRA-2, and JRA-55 (Table 2). The MERIT DEM may
be slightly different than the DEM data used in the three reanalysis products, and this will have a limited impact on
missing data estimation (Sect. 3.3.2).
The Multi-Source Weighted-Ensemble Precipitation (MSWEP) V2.2 dataset (Beck et al., 2017, 2019) is utilized for
the comparison with the SCDNA developed by this study. MSWEP merges data from ground observations, satellite
products, and reanalysis models, and performs better than all products used for merging (Beck et al., 2019). The
comparison can show whether the SCDNA is a better choice than MSWEP to fill gaps in station precipitation
observations.

## 3 Methodology


The methodology to produce the SCDNA includes three primary steps (Fig. 4): (1) preparing a unified precipitation
and temperature database from multiple sources (Sect. 2.1 and 3.1); (2) downscaling reanalysis estimates (Sect. 2.2
and 3.2) that are used in QM- and machine learning-based data estimation (Sect. 3.3) and comparison with the SCDNA
(Sect. 4.5); and (3) producing the SCDNA from 1979 to 2018 based on 16 strategies (Sect. 3.3). The following sub-
sections summarize the work in each step of the methodology (Sect. 3.1, 3.2, and 3.3) as well as the approach used to
evaluate the performance of the method (Sect. 3.4).

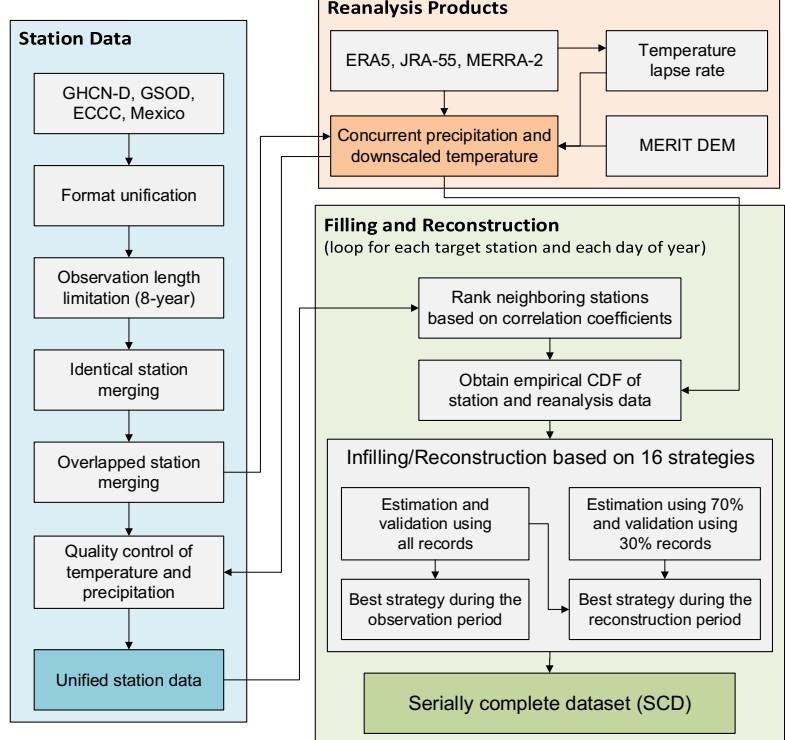


Figure 4. Flowchart of the production of the SCDNA, including station data preparation, reanalysis product processing,
and missing data infilling and reconstruction.
In this study, infilling refers to the estimation of missing values during the observation period, while reconstruction
refers to estimating values outside of the observation period when no station record is available (Fig. 5). Station records
that fail quality control are treated as missing values.

**3.1 Prepare a unified precipitation and temperature database**

*3.1.1 Merging of stations based on location*

Stations are merged if their latitude and longitude match other stations. The problem of overlapped locations is caused
by identification alteration of one station for different periods or recording/rounding bias of station location
information. Although it is possible that multiple stations are deployed in the same location for experimental aims,





location merging is done to preserve internal consistencies as inconsistent records at the same location are self-
contradictory.
The method for location merging includes several steps. First, overlapping stations are extracted and grouped. Stations
within the same group that have non-overlapping recording periods are simply merged into one time series. Otherwise,
the Spearman's rank CC (SCC) between precipitation series from all station pairs in the group is calculated. For SCC
< 0.7, the station group is discarded due to large discrepancies; for 0.7 < SCC < 0.9 the discrepancy is considered as
tolerable and the station with the longest record is kept; for SCC > 0.9 stations are considered as highly correlated and
their data are merged into one time series, while for overlapping periods the station with longest record is used.
Overall, 1240 stations are involved in location merging, stratified in 586 station groups. Around 10% of the groups
contain more than two stations and the largest group contains five stations. After location merging, only 207 groups
are kept and merged into unified times series (Table 1). Despite the steps taken above, the merged series could contain
inhomogeneities due to the combination of records from multiple stations.
*3.1.2 Quality control*
To ensure station observations undergo strict and comprehensive quality control, we adopted the methods used to
produce previous station-based datasets. For $T_{min}$ and $T_{max}$, we followed the method designed by Durre et al. (2010)
which is adopted by GHCN-D (Menne et al., 2012). The procedures include five types of checks: integrity checks,
outlier checks, internal and temporal consistency checks, spatial consistency checks, and extreme megaconsistency
checks. A few of the procedures in Durre et al. (2010) require other variables such as snowfall, and thus are not
adopted in this study. In addition, the quality flags in this study are partly different with those of GHCN-D because of
the different sources, numbers and temporal periods of stations.
For precipitation, quality control procedures consist of three parts. The first part is similar with that for temperature.
The second part (four types of checks) follows procedures designed by Hamada et al. (2011) which are adopted by
the Asian Precipitation-Highly-Resolved Observational Data Integration Towards Evaluation (APHRODITE; Yatagai
et al., 2012). The third part (two types of checks) adopts strategies by Beck et al. (2019) used in the production of
MSWEP. Note that although Durre et al. (2010) and Hamada et al. (2011) share some common traits for precipitation,
both of them are adopted to ensure quality control reliability.
Details of quality checks are in Appendix B.
**3.2 Downscale reanalysis data**
The reanalysis temperature estimates are downscaled to match point-scale station observations using temperature lapse
rate (TLR) according to

$$T_s = T_R + \text{TLR} \times \Delta h \tag{1}$$





where $T_R$ is 2-m reanalysis air temperature, $T_s$ is downscaled temperature, $\Delta h$ is the height difference between station
elevation and reanalysis grid elevation. TLR shows notable spatiotemporal variations (Minder et al., 2010) and
estimating TLR based on ground observations over a large domain is difficult due to the sparsity of stations. Yet recent
studies show that reanalysis outputs offer an alternative in estimating gridded TLR (e.g., Gao et al., 2012). The gradient
of air temperature at different pressure levels above the ground can be used to approximate near-surface TLR (Gao et
al., 2012, 2018; Gruber, 2012). Tang et al. (2018) compared eight temperature downscaling methods in CONUS and
found that methods based on reanalysis-derived TLR can achieve higher accuracy compared to fixed TLR (e.g., -
6.5°C/km) or statistical interpolation downscaling methods. Hence, this study uses the linear regression slope between
MERRA-2 air temperature and geopotential heights from 300 hPa to 1000 hPa pressure levels to represent TLR for
each month at the resolution of 0.5°×0.625° (Table 2). MERRA-2 is used because it directly provides monthly data
and masks temperature data if the pressure level is below land surface. The choice of pressure levels needs further
investigation because relationships between vertical and near-surface temperature vary with regions. Complicated
TLR phenomena such as inverse lapse rate are not considered for simplicity. The climatological mean of TLR (Fig.
S4) decreases from -4.8°C/km in the northeast continent (i.e., Canadian Arctic Archipelago) to -7.2°C/km in the
southwest continent (i.e., Rocky Mountains in CONUS). The smaller TLR magnitude in high latitudes is consistent
with previous studies (e.g., Gardner et al., 2009; Marshall et al., 2007).
**3.3 Produce the serially complete dataset**
To produce the high-quality SCDNA for North America, we use 16 strategies: four based on quantile mapping with
neighboring stations (QMN; e.g., Longman et al., 2019; Newman et al., 2015, 2019), four on quantile mapping with
concurrent reanalysis estimates (QMR), four using spatial interpolation methods (INT; e.g., Eischeid et al., 2000;
Kanda et al., 2018; Woldesenbet et al., 2017), two using machine learning methods (MAL; e.g., Dastorani et al., 2010;
Wambua et al., 2016), and two multi-strategy merging methods (MRG). Merging multiple infilling/reconstruction
methods can provide better estimation than individual methods, as shown by previous data merging and gap infilling
studies (e.g., Eischeid et al., 2000; Beck et al., 2017, 2019; Ma et al., 2018).
We generate estimates for every station and every day from 1979 to 2018 (Fig. 5). The estimates from these 16
strategies and the SCDNA are evaluated using station observations, and the performance of the SCDNA is compared
to four benchmark gridded products. Then, the estimates of the SCDNA are corrected for further accuracy
improvement. Finally, estimates are replaced by station observations when observations exist and pass quality control
checks. The variance and spatial correlation analyses are performed to compare the statistical properties of station
observations and estimates (see Sect. 4).



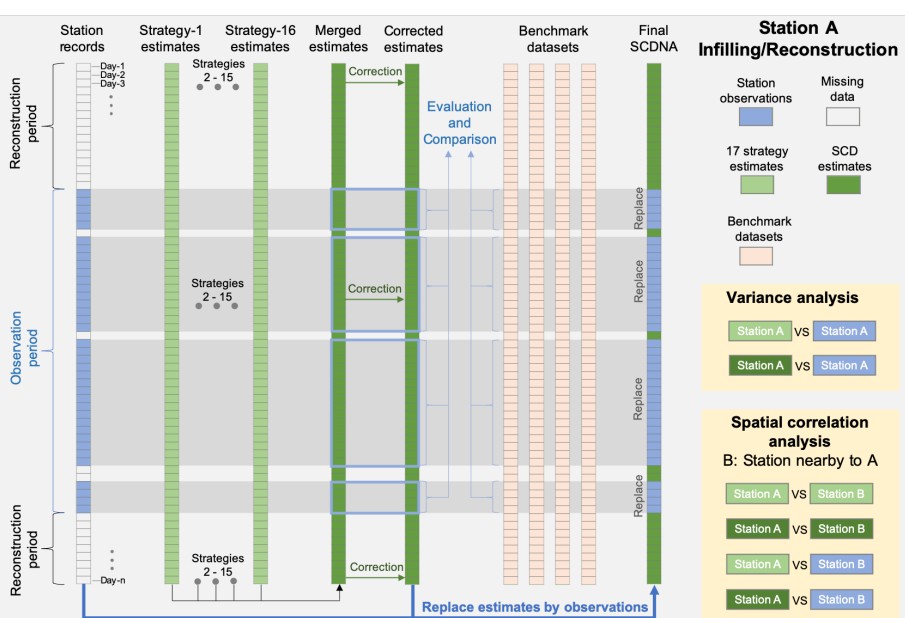

Figure 5. Diagram of the infilling and reconstruction for a specific station (referred to as A). The entire period from 1979 to 2018 is divided into the observation period and the reconstruction period. The data flows of variance and spatial correlation analyses are shown in the nested yellow boxes. Station B is a nearby station of A.

Only stations with at least 3000 valid values are included in the infilling and reconstruction effort. The eight steps (termed Step-1 to Step-8) of SCDNA production are described as below. Unless otherwise stated, the steps are implemented for each target station ($s$), each variable (precipitation, $T_{min}$, and $T_{max}$), and each day of the year (DOY, i.e., 1-366).

*3.3.1 Data extraction*

**Step-1**: Spatiotemporally concurrent reanalysis estimates (ERA5, JRA-55, and MERRA-2) are extracted, including precipitation, $T_{min}$, $T_{max}$, and TLR. Precipitation is linearly interpolated from gridded reanalysis estimates, and temperature is downscaled (i.e., corrected for the elevation difference between the reanalysis grid cell and the station elevation) based on TLR (Sect. 3.1).

**Step-2**: Neighboring stations (at least one and at most 30) with at least 8-year overlapped period with station $s$ are found within the searching radius of 200 km. These stations are ranked from closest to farthest according to their CC with the target station. SCC is used for precipitation, and Pearson CC (PCC) is used for $T_{min}$ and $T_{max}$. CC is calculated using data within a 31-day window centered around the current DOY from all years.

**Step-3**: The empirical CDFs of $s$, neighboring stations, and reanalysis estimates are obtained using data within the same 31-day window.



*3.3.2 Infilling and reconstruction*
**Step-4**: For each day (*d*) corresponding to the DOY, the estimated data are acquired based on 16 strategies which are
divided into five groups.
*Group 1*: **Quantile Mapping with Neighboring stations (QMN)**
•    **QMN-1**: For all neighboring stations with valid records, the station with the highest CC in Step-2 is selected.

The estimated data for *s* and *d* is obtained using Eq. (2).

$$X_s = F_s^{-1}(F_i(X_i)) \qquad (2)$$

where $X_i$ is precipitation or temperature for *d* from the selected neighboring station *i*, $F_i$ is the empirical CDF of
*i* corresponding to the DOY, $F_s^{-1}$ is the inverse CDF of *s* corresponding to the DOY, and $X_s$ is the estimated data.
•    **QMN-2**: For all neighboring stations with observations, estimated values are obtained using Eq. (2) which are

merged based on Eq. (3).

$$X_s = \frac{\sum_i^n W_i F_s^{-1}(F_i(X_i))}{\sum_i^n W_i} \qquad (3)$$

$$W_i = CC_i^2 \qquad (4)$$

where *n* is the number of neighboring stations, $F_s^{-1}(F_i(X_i))$ is the QM-based estimate from *i*, and $W_i$ is the weight
calculated using Eq. (4). $CC_i$ is CC (SCC or PCC) between data from *s* and *i* corresponding to the DOY. $W_i$ is
assigned zero if $CC_i$ is negative.
•    **QMN-3**: Similar to QMN-2, but the weight is calculated according to the distance ($D_i$) between *s* and *i* based on

Eq. (5). Although the exponent of distance (*k*) varies in different studies, -2 is the most common choice

(Teegavarapu and Chandramouli, 2005).

$$W_i = D_i^k \qquad (5)$$

•    **QMN-4**: The median of QMN-1 to QMN-3 is used as the estimated data. The strategy of using median values is

the same with Eischeid et al (2000), which could be closer to actual observations than QMN-1 to 3.

*Group 2*: **Quantile Mapping with Reanalysis products (QMR)**
Reanalysis products provide useful information for SCDNA production as (1) remote regions may not have enough
neighboring stations, and (2) neighboring stations also have missing values which could result in gaps of estimates at
the target station.





•    **QMR-1 to QMR-3**: Similar to QMN-1, but the neighboring station is replaced by concurrent ERA5, JRA-55,
and MERRA-2 estimates, respectively.

•    **QMR-4**: The median of QMR-1 to 3 is used as the estimated data.
*Group 3*: **Interpolation (INT)**
The three interpolation methods used in this study are MLAD (referred as INT-1), NR (referred as INT-2), and inverse
distance weighting (IDW, referred as INT-3). They are described below. Following Eischeid et al. (2000), neighboring
stations with CC lower than 0.35 are excluded. The remaining stations are ranked from high CC to low CC. A
maximum of four neighboring stations are used in the interpolation. For $T_{min}$ and $T_{max}$, direct interpolation from
neighboring stations to $s$ could be biased due to the elevation differences between stations. Temperature data from
neighboring stations are downscaled to the elevation of $s$ based on Eq. (1).
•    **INT-1**: MLAD minimizes the sum of absolute errors. It is more robust than regression based on least squares
because while least square estimation is effective when the errors are normally distributed and independent,
environmental variables, especially precipitation, often violate the assumption of normality (Eischeid et al.,
2000). MLAD has been well documented with better performance in gap infilling than other interpolation
methods (Eischeid et al., 1995, 2000; Kanda et al., 2018; Young, 1992). The formula is shown in Eq. (6).

$$X_s = c_0 + \sum_{i}^{n} c_i X_i \qquad (6)$$

where $c_i$ ($i = 0, 1, …, n$) is regression coefficients estimated using data within a 31-day window for each DOY.
Different $d$ corresponding to the same DOY could have different combinations of neighboring stations due to the
limitation of observation availability. MLAD is performed for each combination to ensure that effective estimates
are available for all days.

•    **INT-2:** NR is an interpolation method proposed by Paulhus and Kohler (1952) and modified by Young (1992).
The modified version is adopted in this study, which combines information from neighboring stations by
replacing $F_s^{-1}(F_i(X_i))$ with $X_i$ in Eq. (3). The weight is calculated using Eq. (7).

$$W_i = CC_i^2 \frac{N_i - 2}{1 - CC_i^2} \qquad (7)$$

where $N_i$ is the number of samples used to calculate $CC_i$ between $s$ and $i$. SCC is used for precipitation and PCC
is used for temperature.

•    **INT-3**: IDW is one of the most common interpolation methods. It is implemented similar to NR, where the
inverse squared distance, as shown in Eq. (5), is used as the weight.



•   **INT-4**: The median of INT1, INT2 and INT3 is used as the estimated data.
*Group 4*: **Machine Learning (MAL)**
The two MAL methods used in this study are ANN (referred as MAL-1) and random forest (RF, referred as MAL-2;
Breiman, 2001). Unlike QMN, QMR and INT that are carried out for each DOY, MAL uses complete observation
records of $s$ to ensure that ANN and RF are trained with enough values. MAL models are trained using the first 70%
observations and tested using the remaining 30% observations. The MAL models' validation based on the 30%
observations can indicate their performance in the reconstruction period.
The input data are from neighboring stations and concurrent reanalysis estimates. For each $s$, neighboring stations are
determined in a way similar with Step-2, but CC is calculated using data in the entire observation period. Neighboring
stations with CC lower than all reanalysis products (ERA5, JRA-55, and MERRA-2) are excluded. The remaining
neighboring stations and three reanalysis products form a complete repository of input features. Then, for each day
that $s$ has no observation, the input features are extracted from the repository in three steps: (1) neighboring stations
without observations for the day are excluded, (2) the remaining neighboring stations and reanalysis products are
ranked according to their CC with $s$, and (3) at most five stations/reanalysis products with the highest CC are selected.
In this way, $s$ will have multiple combinations of input features to ensure that all days with missing values have
estimates. All combinations are used to train and test the ANN and RF models, resulting in multiple estimated series
for $s$. The final estimates of $s$ are generated in three steps: (1) the Kling-Gupta Efficiency (KGE; Kling et al., 2012)
of all estimated series is calculated using all observations of $s$, and ranked from high to low KGE (see Sect. 3.4 for
definition of KGE); (2) the series with higher KGE is used to constitute the estimates of $s$ in sequence; and (3) the
second step is repeated until there are no missing values for $s$. This approach ensures that "best" and complete estimates
are provided for $s$.
•   **MAL-1**: A four-layer ANN is used. The input layer has a maximum of five nodes (depending on the number of
input features), the two hidden layers both have 20 nodes, and the output layer has one node for generating
precipitation or temperature estimates. The transfer functions are hyperbolic tangent sigmoid for hidden layers
and linear for the output layer. The training function is resilient backpropagation. The model is trained using the
first 50% data, validated using the subsequent 20% data, and tested using the final 30% data.

•   **MAL-2**: A RF model with 50 trees is built with 70% training data and 30% testing data. The minimum number
of samples per tree leaf is 5. The input nodes depend on the number of input features like MAL-1.

*Group 5*: **Multi-Strategy Merging (MRG)**

•   **MRG-1**: KGE is used to rank the performance of the 11 strategies (QMN-1 to 3, QMR-1 to 3, INT-1 to 3,
and MAL-1 to 2) as CC cannot reflect the magnitude difference (e.g., bias) between target and reference



series. The first three cases of the 11 strategies are merged using squared KGE as the weight. The individual weight is assigned zero if KGE is negative.

- **MRG-2**: The median of the three selected strategies in MRG-1 is used as the estimated data.

*3.3.3 Generating serially complete records*

**Step-5**: In this step, Step-3 and -4 are repeated based on 70% data of $s$ in the observation period. Then, the KGE of estimates from all strategies are calculated using the remaining 30% observations. MAL-1 and 2 are not repeated because they are trained on the 70% observations. This step is implemented because QMN-1 to 4, QMR-1 to 4, and INT-1 in Step-4 use all data of $s$ in the observation period to select stations, estimate empirical CDFs and carry out regression. This potential overfitting problem could lead to better performance of these strategies in the observation period but worse performance in the reconstruction period. KGE calculated in Step-4 can represent the accuracy of estimates in the observation period, while KGE calculated in Step-5 can represent the accuracy of estimates in the reconstruction period.

**Step-6**: In the observation period, the strategy with the highest KGE in Step-4 is selected to contribute the extension/reconstruction to the SCDNA. In the reconstruction period, first, the strategy with the highest KGE in Step-5 is determined; then, the estimates from the corresponding strategy in Step-4 are used to constitute the SCDNA because the empirical CDF and regression based on all observations in Step-4 could be more representative than the 70% observations in Step-5.

**Step-7**: Estimates in Step-6 are corrected for certain climatological biases using station data in the observation period. Precipitation estimates are often subjected to wet-day bias. Two methods are implemented to address this problem. First, QM is performed based on the CDF of $s$ in Step-3. However, QM may reduce the accuracy of estimated precipitation in some cases, for which the method used in Beck et al. (2019) is adopted. This method subtracts a tiny value (0.01 mm) from the original precipitation series and rescales the series to restore the original mean value. This operation is repeated until the estimated series show equal number of wet days (>0.5 mm d$^{-1}$) with observations of $s$. In addition to wet-day bias correction, mean-value correction is implemented. The ratio between the mean values of precipitation estimates and observations is calculated in the observation period, which is used to rescale estimated series in both observation and reconstruction periods. For $T_{min}$ and $T_{max}$, QM correction and mean-value correction are also implemented.

**Step-8**: The accuracy of the SCDNA is evaluated and compared to benchmark datasets based on actual observations (Fig. 5). Then, the estimates are replaced by observations whenever possible to generate the final SCDNA. Very occasionally, estimated $T_{min}$ could be larger than estimated $T_{max}$, for which $T_{max}$ is replaced by the maximum $T_{max}$, and $T_{min}$ is replaced by the minimum $T_{min}$ of the estimates from the 16 strategies.





**3.4 Evaluate the precipitation and temperature estimates**

KGE, which is proposed by Gupta et al. (2009) and modified by Kling et al. (2012), is used to support the merging of different strategies (Sect. 3.3) and the evaluation of the estimated precipitation and temperature. It is a useful metric in evaluating various variables (e.g., Tang et al., 2020) and incorporates information about correlation, bias, and variance.

$$\begin{cases} \text{KGE} = 1 - \sqrt{(r-1)^2 + (\beta-1)^2 + (\gamma-1)^2} \\ \beta = \dfrac{\mu_s}{\mu_0} \\ \gamma = \dfrac{CV_s}{CV_o} = \dfrac{\sigma_s/\mu_s}{\sigma_o/\mu_o} \end{cases} \tag{8}$$

where $r$ is the PCC, $\beta$ is the bias ratio, and $\gamma$ is the variability ratio; $\mu$ is the mean value, and $\sigma$ is the standard deviation. The subscripts $s$ and $o$ represent estimated and reference time series, respectively. KGE ranges from negative infinity to one. If two series exactly match, the KGE is one. A $\beta$ or $\gamma$ value smaller/larger than one indicates that the mean value or variability of observations is underestimated/overestimated.

In Sect. 4, the evaluation during the observation period is based on the complete station observations (i.e., Step-4 in Sect. 3.3.2), while the evaluation during the reconstruction period is realized using 30% independent station observations (i.e., Step-5 in Sect. 3.3.3). Unless otherwise stated, SCDNA estimates in Sect. 4 are after correction (Step-7 in Sect. 3.3.3). In Sect. 4.5, SCDNA estimates are compared with gridded products (ERA5, JRA-55, MERRA-2, and MSWEP). In addition to the three SCDNA variables (precipitation, $T_{min}$, and $T_{max}$), mean temperature ($T_{mean}$, the mean of $T_{min}$ and $T_{max}$) and daily temperature range ($T_{range}$, the difference between $T_{max}$ and $T_{min}$) are also included. The involvement of $T_{range}$ can contribute to more objective comparison between SCDNA and reanalysis products because the TLR-based downscaling of reanalysis temperature contains uncertainties, which could affect the evaluation of $T_{min}$, $T_{max}$, and $T_{mean}$. Although there exist differences between TLR of $T_{min}$ and $T_{max}$, $T_{range}$ can reduce the effect of scale-mismatch between gridded reanalysis temperature and point station temperature on evaluation results.

**4 Results**

**4.1 Comparison of infilling and reconstruction strategies**

The value of a given infilling/reconstruction strategy can be quantified by the extent that a strategy is selected for use in the final SCDNA dataset. In this sense the contribution ratios define the proportion of estimates that come from a specific strategy. Fig. 6 shows that the contribution ratios of QMN, QMR, and INT to missing value estimation are generally smaller than 20% in North America. Please note that QMN refers to all strategies within this group unless the strategy number is specified right after QMN. This also applies to other groups. QMR shows the smallest contribution ratios for almost all stations among the five groups. Compared with other regions in North America, contribution ratios of QMR are higher for precipitation stations in western U.S. and temperature stations in Mexico.

INT shows lower contribution ratios in Rocky Mountains compared with western U.S., indicating statistical
interpolation without considering topographic effect is subjected to substantial uncertainties in complex terrain. MAL
shows notably higher contribution ratios than QMN, QMR, and INT, particularly for $T_{min}$ and $T_{max}$. The ratios of MAL
are higher than 20% for ~30% precipitation stations, ~65% $T_{min}$ stations, and ~68% $T_{max}$ stations. MRG has the highest
contribution ratios throughout North America. The average contribution ratios of MRG are 59.88%, 41.59%, and
40.56% for precipitation, $T_{min}$, and $T_{max}$, respectively. For precipitation, MRG is particularly effective in high-latitude
regions (northern Canada and Alaska), western U.S. and Mexico.

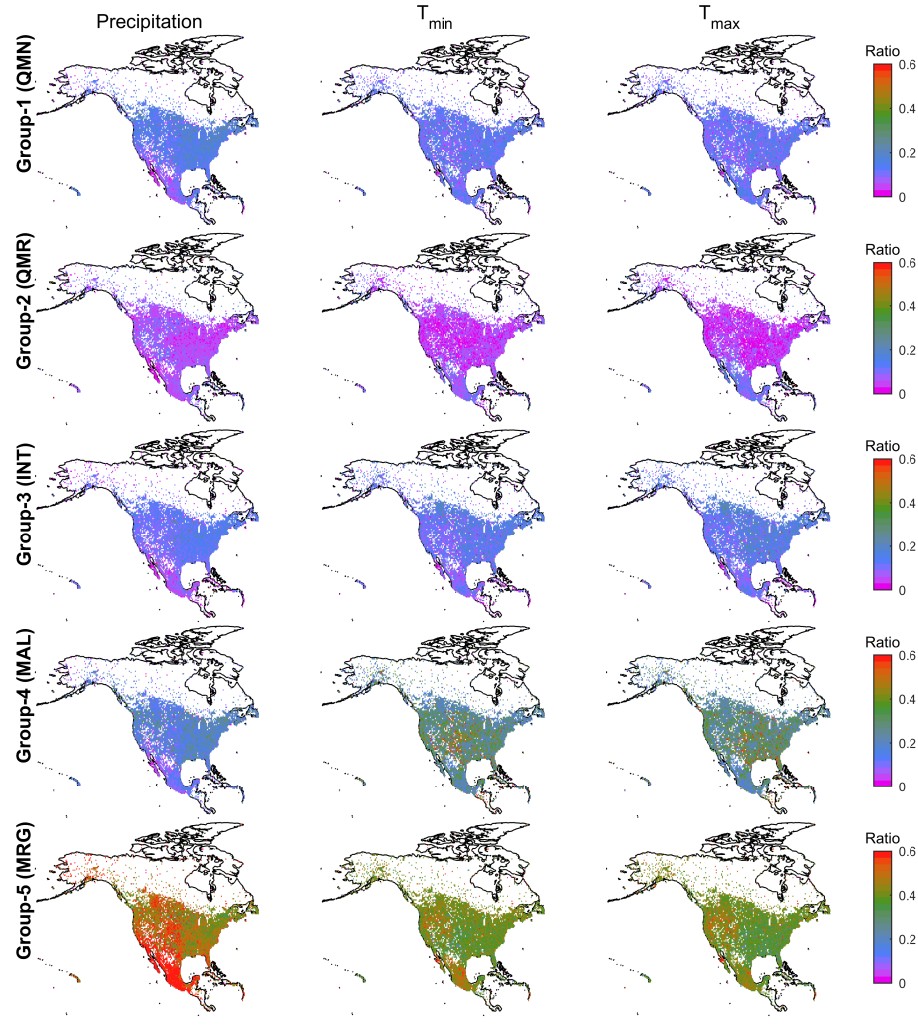


Figure 6. The contribution ratios of estimates from five infilling/reconstruction groups to the missing values of all
stations from 1979 to 2018. The three columns from left to right represent precipitation, $T_{min}$, and $T_{max}$, respectively.





The five rows from top to bottom represent Group-1 (QMN), Group-2 (QMR), Group-3 (INT), Group-4 (MAL), and
Group-5 (MRG), respectively. The maps are at the resolution of 0.5°. The ratio for each grid cell is the mean value of
all stations within this grid cell.
Figure 7 shows the KGE and contribution ratios of 16 strategies. The KGE of estimated precipitation is lower than
that of estimated $T_{min}$ and $T_{max}$ due to the stronger spatial and temporal homogeneity of temperature (Fig. 7). The
median KGE values of $T_{min}$ and $T_{max}$ are generally above 0.9, and the accuracy of estimated $T_{max}$ is higher than that of
$T_{min}$. The KGE during the reconstruction period is smaller than that during the observation period, which is particularly
obvious for QMN, QMR, and INT-1 compared with other strategies, because QMN and QMR transfer CDF during
the observation period to other periods, and INT-1 transfers regression relationship during the observation period to
other periods. MAL suffers a slight degradation in the reconstruction period, and the better performance of MAL-2
than MAL-1 shows that RF could be a better choice than ANN in estimating missing data. For MRG, the differences
of KGE between the two periods are relatively small. For example, the median KGE values of MRG-1 for $T_{max}$ are
0.99 and 0.98 for observation and reconstruction periods, respectively. MRG also shows higher KGE and a narrower
quantile ranges than other strategies, particularly for precipitation, benefiting from merging estimates from multiple
strategies
Regarding contribution ratios (Fig. 7), strategies with higher KGE often have larger contributions to the estimated
series. However, this is not always true because the selection of strategies is performed for each DOY. Note that the
contribution ratios of MAL-2 are even higher than MRG-1 during the observation period for $T_{min}$ and $T_{max}$, although
MRG-1 achieves higher KGE than MAL-2 for most stations. This is because MAL-2 could be the best choice for
more DOY than MRG-1 even though MRG-1 may achieve the best overall performance. An example using $T_{min}$ data
from one station is shown in Fig. S5.
In the reconstruction period when observations are absent, the contribution ratios of MAL-2 decrease drastically
compared with the observation period, contributing to the increased ratios of other strategies (particularly MRG-1).
Although QMR shows the lowest contribution ratios, reanalysis products have implicit contributions to other strategies
(e.g., MAL and MRG). Overall, MRG-1 shows much higher contribution ratios than all the other strategies (including
MRG-2) during the reconstruction periods, indicating that it is the most important strategy in missing value estimation.
Hence, combining information from multiple strategies is more reliable, and KGE-based merging is more effective
than the median-value-based estimation.



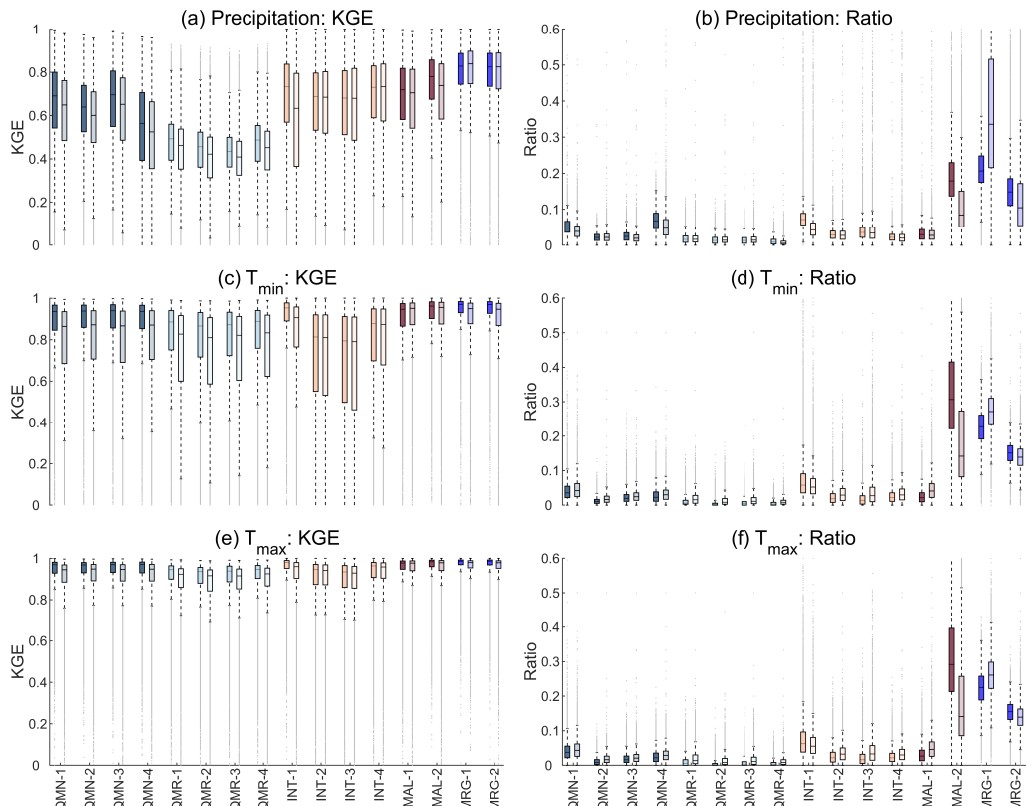

Figure 7. Boxplots of (a, c, and e) the KGE and (b, d, and f) the contribution ratio of 16 strategies for all stations. Each strategy corresponds to two boxes in each sub-figure; the left one with darker color represents the observation period, and the right one with lighter color represents the reconstruction period. The line within the box is the median. The upper and lower edges of the box represent the 25th and 75th percentiles, respectively. Values more than 1.5 times the interquartile range away from the upper or lower edges are outliers (dots).

**4.2 Impact of reconstruction on spatial correlation and series variance**

All infilling/reconstruction strategies except QMR rely on information from neighboring stations; this could affect the spatial correlation structure and the variance of SCDNA series. Space-time correlations and other properties (e.g., intermittency of precipitation) are important considerations because they can influence the performance of follow-on applications that use the SCDNA as input. Theoretically, QMN strategies could significantly inflate spatial correlation but retain variance of station observations. The spatial correlation inflation in INT strategies could be lower but the variance would be underestimated due to smoothing. QMR-1 is used as an example to demonstrate the effect of QM on spatial correlation and series variance (Fig. S6), because QMN uses different station combinations for every DOY which would mask the effect of QM on final estimates. If the ERA5 used by QMR-1 is replaced by station observations, the results should be generally consistent. According to Fig. S6, the spatial correlation is substantially inflated by



QMR-1, particularly for $T_{min}$ and $T_{max}$, while the standard deviation of QMR-1 estimates is very close to that of
observations. This supports the design of estimating missing data using neighboring stations for each DOY as
otherwise the inflation of CC could be very substantial for the entire period.
The spatial correlation based on station observations (Fig. 8a, d, and g) shows obvious seasonal variations, with CC
lower in the warm season and higher in the cold season. The seasonality of CC for $T_{max}$ is weaker compared with that
for precipitation and $T_{min}$. The SCDNA estimates capture the seasonal patterns but underestimates the variation (Fig.
8b, e, and h) because the inflation of spatial CC is larger in the warm season than cold season (Fig. 8c, f, and i).
Moreover, the inflation is larger for neighboring stations with lower correlation with the target station. We tested
selecting neighboring stations according to their distance from the target station, and similar results were acquired.
For precipitation, the median CC differences of all stations are close to 0.1 in the cold season and ranges between 0.1
and 0.15 in the warm season. For $T_{min}$, the median CC differences are generally between 0.05 and 0.15. The CC
differences of $T_{max}$ are relatively homogeneous for different seasons and generally fluctuate between 0.05 and 0.1. The
inflation of CC is because (1) the estimates from the 10 neighboring stations and the target station are generally derived
using highly overlapped information (Sect. 3.3.1), and (2) estimation is realized for each DOY for all strategies except
MAL, meaning that calculating CC for each DOY show the inflation to the largest extent.
The final SCDNA replaces estimates by observations, which can largely relieve the inflation of spatial correlation
(Fig. S7), depending on the degree to which observations are present in the record. For $T_{min}$ and $T_{max}$, CC is very close
to that based on observations; for precipitation, correlation in wintertime is even lower than that based on observations.

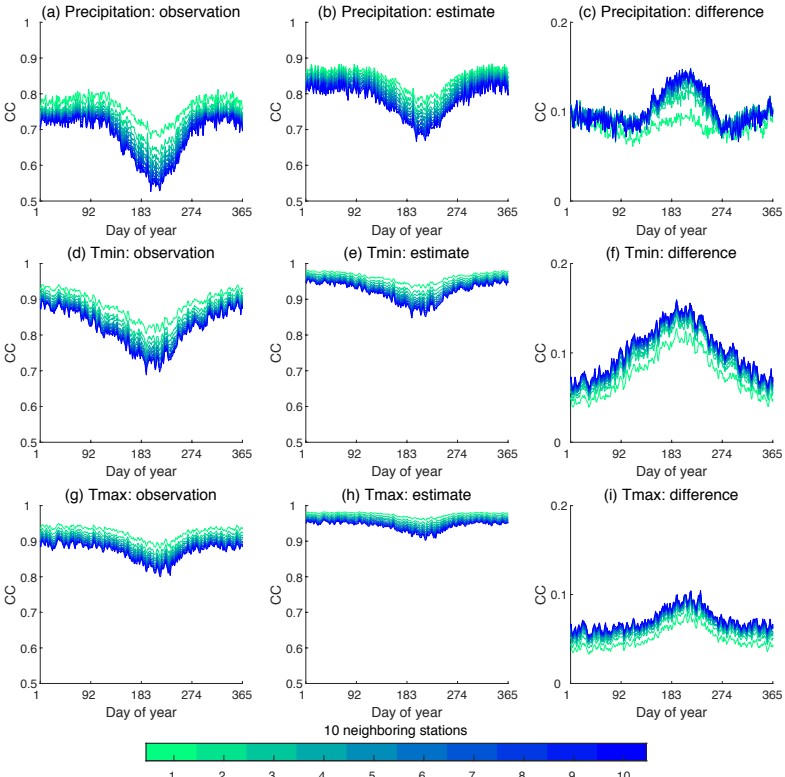


Figure 8. CC between target and neighboring stations for all DOY using station observations (the first column), SCDNA estimates (second column), and differences between SCDNA- and observation-based CC (the third column). CC is calculated in the observation period. For each target station, 10 neighboring stations are selected according to the correlation between time series from target and neighboring stations. Smaller numbers represent higher correlation. For example, station 1 represents the neighbor with the highest CC with the target station. Each curve represents the median CC of all stations.

Figures 9 and 10 show CC between estimates at the target station and observations at the neighboring station. For
precipitation, most strategies exhibit similar spatial correlation structure with observations for most stations. QMR
largely underestimates CC compared with observations, which should be attributed to the differences between
precipitation of reanalysis products and stations. There are notable differences for different strategies within one group.
For example, QMN-1 shows larger inflation when observation-based CC is higher, which is not seen in QMN-2 to 4.
This is probably because QMN-1 only uses information from the one neighboring station with the highest correlation
with the target station for each DOY. Higher observation-based CC in Fig. 9 means this neighboring station could be
more frequently used by QMN-1 to estimate data for the target station, resulting in the larger inflation of CC. Another
example is that INT-1 underestimates the CC for 68.75% stations, whereas INT-2 to 4 overestimates the CC for almost
all stations. For SCD-1, inflation of CC is observed for 76.60% stations, whereas the magnitude of overestimation is



smaller than that in Fig. 8. The mean values of observation-based and estimate-based CC are 0.71 and 0.77, respectively. SCD-2 replaces estimates by observations and is the final dataset of this study. It reduces the mean estimate-based CC to 0.70. The overall spatial correlation structure of observations is generally preserved by SCD-2. However, SCD-2 calculates CC for the entire period which is different from the period of observation-based CC, resulting in uncertainties such as the underestimation for some stations when observation-based CC is larger than 0.7.

The spatial correlation of $T_{min}$ is much stronger than that of precipitation (Fig. 10). Most strategies overestimate the CC for most stations, whereas the magnitude is quite small. For example, SCD-1 inflates the CC for 96.96% stations, while the mean CC values for observations (0.95) and SCD-1 (0.96) are very close to each other. QMR still underestimates CC similar to Fig. 9 for precipitation. CC based on SCD-2 is generally consistent with that based on observations, while slight underestimation exists for some stations when observation-based CC is higher than 0.9. $T_{max}$ shows similar spatial correlation patterns with $T_{min}$ (Fig. S8).

In summary, inflation of CC is inevitable particularly when estimates are obtained using information from sole data source such as one neighboring station or one reanalysis product. The inflation is larger if each DOY is treated separately (Fig. 8 and S7), but smaller if CC is calculated for all years (Fig. 9, 10 and S8). Combining information from multiple sources (stations and reanalysis) and combining multiple strategies for each DOY are beneficial in estimating the overall spatial correlation structure. The spatial correlation structures vary for different strategies, and further studies are needed to clearly demonstrate how and why the estimate-based CC differs from observation-based CC.

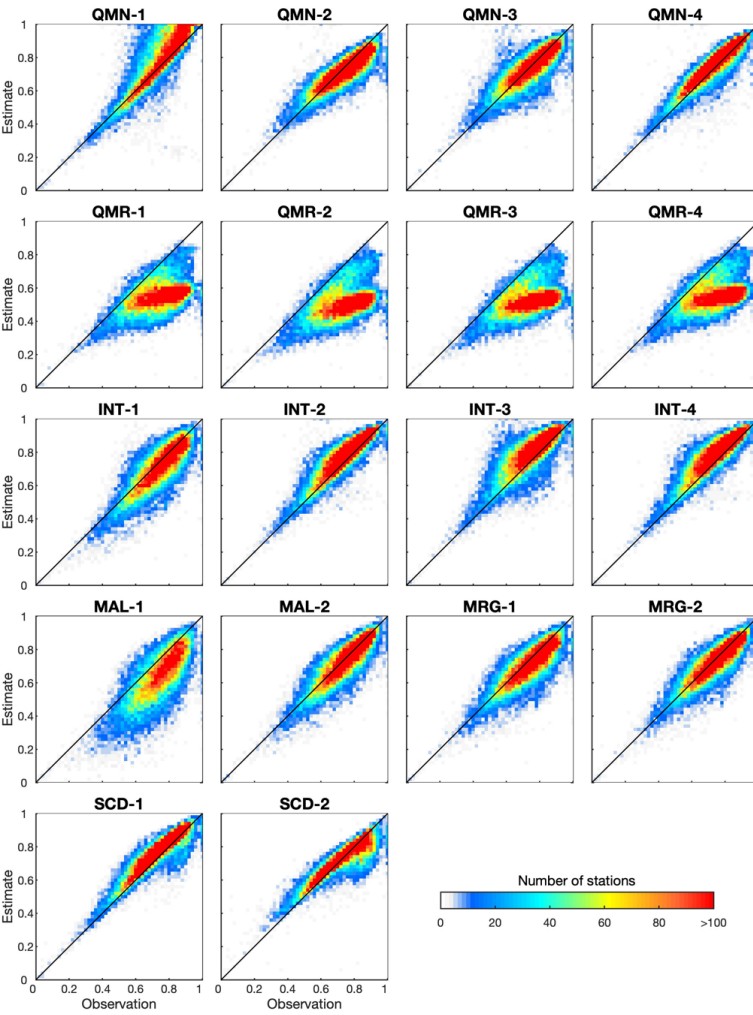

Figure 9. Scatter density plots of CC between precipitation from the target station and neighboring stations. For each target station, the neighboring station with the highest correlation with the target station is selected. X-axis represents the CC between observed precipitation from target and neighboring stations. Y-axis represents the CC between estimated precipitation from the target station and the observed precipitation from the neighboring station. Each sub-figure corresponds to one strategy in Sect. 3.3.2. SCD-1 represents SCD estimates after correction, while SCD-2 replaces estimates by observations. CC is calculated during the overlapped observation period between target and neighboring stations, and the only exception is SCD-2 which calculates CC using precipitation from target and neighboring stations during the entire period.

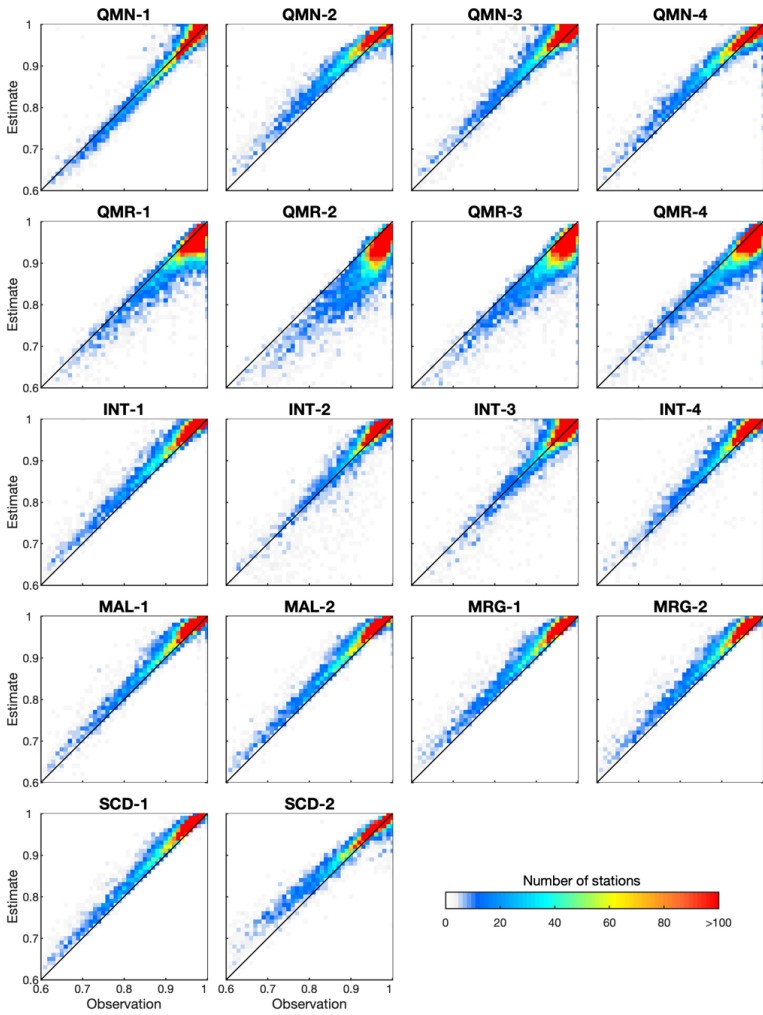

Figure 10. Similar with Fig. 9, but for $T_{min}$.

The variability of observations and of the corrected and uncorrected SCDNA estimates (Step-7 in Sect. 3.3.3) are
compared using the standard deviation of the observation period (Fig. 11). The standard deviation of uncorrected
SCDNA precipitation is lower than that of observations, while after correction, the standard deviation agrees very well
with observations. The mean values of standard deviation are 7.36, 6.30, and 7.36 for observations, uncorrected
SCDNA, and corrected SCDNA, respectively. For $T_{min}$ and $T_{max}$, corrected and uncorrected SCDNA estimates both
show consistent variability with observations.



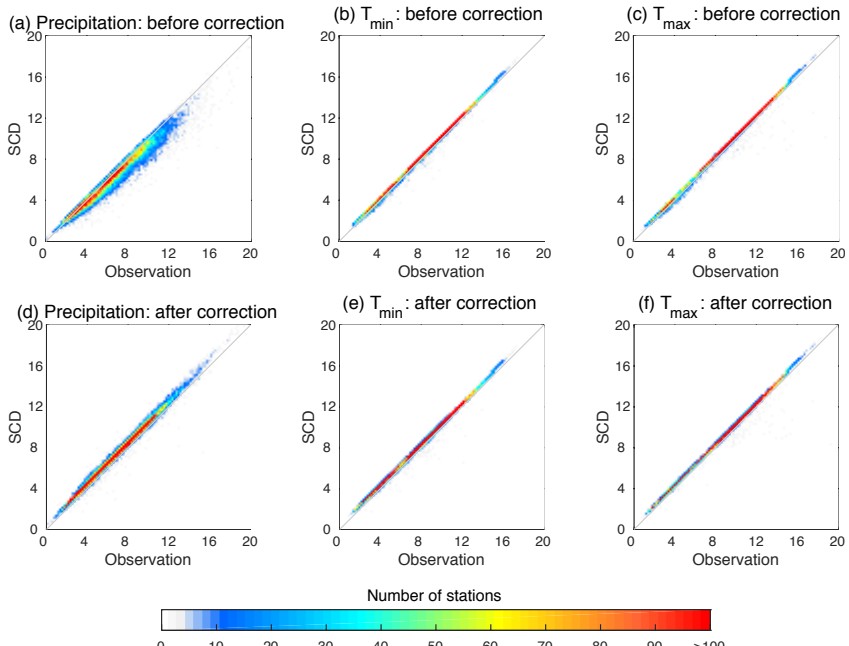

Figure 11. The standard deviation of observations and SCDNA estimates before and after correction. Data in the observation period are used.

**4.3 The performance of the serially complete dataset**

Uncorrected SCDNA estimates show high accuracy in North America (Fig. 12). For precipitation, the median KGE of all stations is 0.87, and the median values of $r$, $\beta$, and $\gamma$ are 0.91, 0.92, and 0.96, respectively. The KGE for Mexico stations generally ranges between 0.6 and 0.8, which is smaller than that in U.S. and southern Canada. Some stations in Rocky Mountains, Caribbean, Alaska and northern Canada (regions with complex topography or climate), also show lower KGE for precipitation estimates. The spatial distribution of $r$ is similar with that of KGE, while the magnitude is higher. According to $\gamma$, most stations underestimate precipitation variability which is consistent with Fig. 11; $\beta$ is generally lower than one in most regions of North America, particularly in Rocky Mountains and Mexico where SCDNA underestimates precipitation.

Estimated temperature shows much higher KGE compared with precipitation. The median KGE and $r$ of $T_{min}$ are 0.97 and 0.99, respectively. For $T_{max}$, the median of KGE and $r$ are 0.99 and 0.99, respectively. The median $\gamma$ and $\beta$ are both between 0.99 and 1 for $T_{min}$ and $T_{max}$ with small variations, particularly for $T_{max}$ (Fig. 12); the KGE of $T_{min}$ and $T_{max}$ is lower in Caribbean and Mexico. For $T_{min}$, the KGE for some stations around 45°N and Rocky Mountains is lower than surrounding regions although $\gamma$ is spatially homogeneous for the same region. $T_{max}$ exhibits homogeneous performance in the same region for all metrics. The discrepancies between $T_{min}$ and $T_{max}$ need further investigation.

Corrected SCDNA estimates (see Step-7; Fig. S9) have higher accuracy than uncorrected estimates (Fig. 12). For
example, the median KGE for precipitation is improved from 0.87 to 0.90 after correction. The KGE for $T_{min}$ and $T_{max}$
is also improved but not as significant as precipitation. $\beta$ equals to one for all stations due to the mean-value correction.
$\gamma$ for precipitation changes from negative to positive for all stations, whereas magnitude of bias (deviation from one)
is smaller after correction. The spatial distribution of metrics for $T_{min}$ is also more homogeneous. Therefore, the
correction procedures are effective.

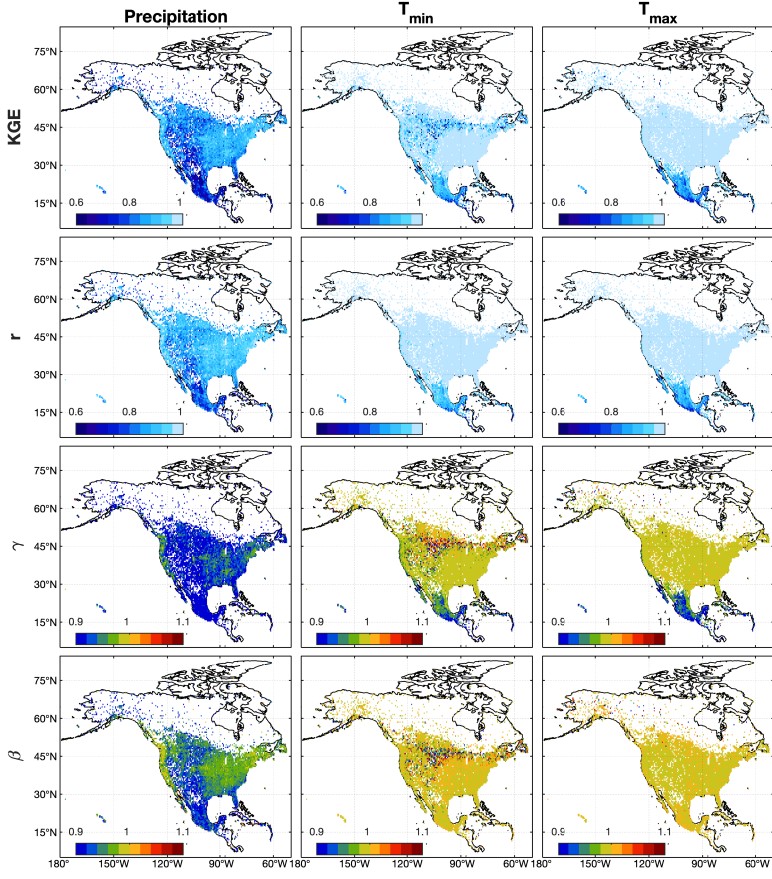


Figure 12. The spatial distributions of KGE and its three components ($r$ is CC, $\beta$ is the bias ratio, and $\gamma$ is the
variability ratio) for uncorrected SCDNA estimates over North America during the observation period. The maps are
at the resolution of 0.5°. The value for each grid cell is the median value of all stations within this grid cell.
The distributions of KGE vary during the year (Fig. 13). For precipitation, more stations show lower KGE during
summer (DOY 150 to 250) than at other times of the year, which may be due to the variability of summertime
convective precipitation. For $T_{min}$, some stations show lower KGE from DOY 100 to 250. The seasonal variation of



KGE for $T_{max}$ is relatively weak, although KGE is slightly more concentred at higher level during spring and autumn
than winter and summer. The overall performance of $T_{max}$ is better than $T_{min}$ and precipitation.

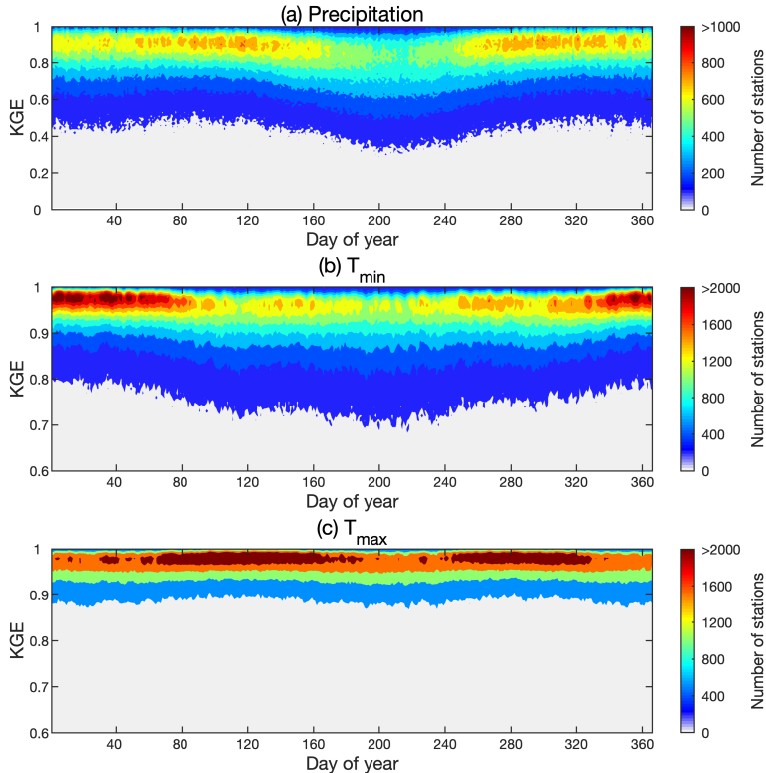


Figure 13. The distribution of KGE for each day of year for (a) precipitation, (b) $T_{min}$, and (c) $T_{max}$. Corrected SCDNA
estimates are used.

### 4.4 Comparison between the serially complete dataset and gridded products

SCDNA precipitation and temperature are compared with benchmark gridded products to demonstrate whether the
SCDNA is a good choice when station data are unavailable. Actual station observations are used as reference.
Although assessing gridded products using point-scale station data contains uncertainties (Tang et al., 2018a), the
objective of this section is to illustrate their agreement with station observations in lieu of provide an exhaustive
quantitative assessment of their real-world accuracy.
Overall, the SCDNA achieves much higher KGE than reanalysis products for all variables (Fig. 14). For precipitation,
the median KGE differences between the SCDNA and ERA5, JRA-55 and MERRA-2 are 0.48, 0.57, and 0.54,
respectively. The corresponding KGE differences for $T_{min}$ are 0.46, 0.61, and 0.36, respectively. The improvement for
$T_{max}$ is smaller, particularly in eastern U.S. where topography is relatively flatter compared with western U.S. The
KGE differences of $T_{mean}$ are lower than $T_{min}$ but higher than $T_{max}$ due to the offset effect. $T_{range}$ suffers little from the
elevation differences between stations and reanalysis grids, and is suitable to demonstrate the differences between
SCDNA and reanalysis products. The median KGE differences for $T_{range}$ between the SCDNA and ERA5, JRA-55 and
MERRA-2 are 0.31, 0.48, and 0.31, respectively.

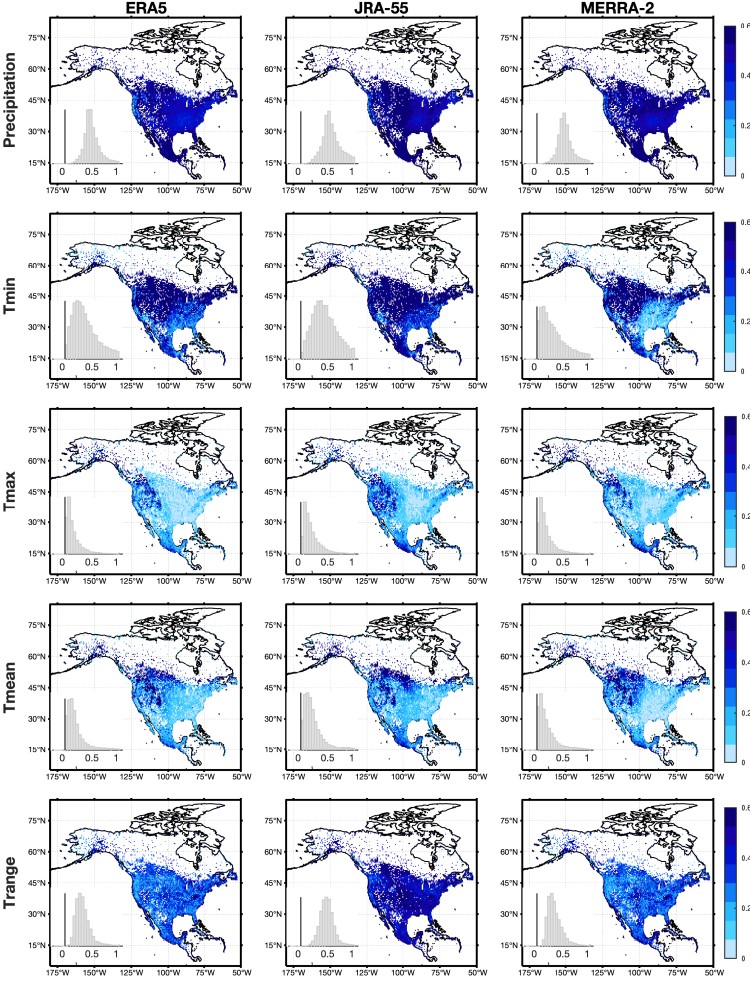


Figure 14. Spatial distributions of KGE differences between SCDNA estimates and three reanalysis products (ERA5,
JRA-55, and MERRA-2). The nested histograms show KGE differences between the SCDNA and reanalysis products.
Corrected SCDNA estimates are used.
SCDNA and MSWEP precipitation is compared (Fig. 15). Since MSWEP merges data from numerous stations, the
evaluation of MSWEP based on station data is not independent, which could result in the overestimation of its KGE.
Even so, SCDNA precipitation shows higher KGE than MSWEP for 98.97% stations with a median KGE difference





of 0.31. Fig. 15 shows notable differences between Canada, U.S. and Mexico which could be due to the differences
in observation time of stations in different countries. The accumulation periods of station and MSWEP precipitation
are inconsistent in some cases, which could affect the evaluation of MSWEP (see Sect. 5.1).
Note that the evaluation does not indicate that the SCDNA has higher accuracy than the gridded products; rather, the
results show that SCDNA is a better substitute than gridded products when station observations are unavailable.

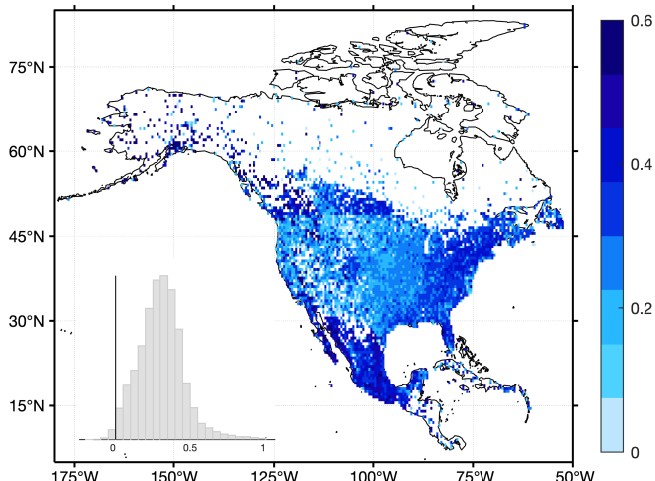


Figure 15. Spatial distributions of KGE differences between SCDNA and MSWEP precipitation. Corrected SCDNA
estimates are used.
**5. Discussion**
**5.1 Observation time of stations**
Meteorological stations in different countries usually have different local observation time, and stations in the same
country may also experience change of observation time (Vincent et al., 2012). Most station databases including those
used in this study do not account for reporting-time inconsistencies due to lack of hourly observations and well-
documented station metadata. Vincent et al. (2009) examined several methods to adjust the time of daily precipitation
observations, which, however, often altered observed precipitation intensity. Beck et al. (2019) inferred the reporting
time of daily precipitation observations by calculating SCC between the series of stations and gridded products, which
is useful to correct the bias of gridded products. A simple experiment is carried out using the method of Beck et al.
(2019) to infer the lag day of station series. For precipitation, 6418 stations show nonnegligible time shift from the
reporting date (Fig. S10). However, this method may be unsuitable for temperature because the estimated lag day is
mostly zero, and the inferred reporting time cannot be directly applied to adjust station observations.



The inconsistent reporting time has different impact on precipitation, $T_{min}$, and $T_{max}$. For example, if a station records
data from 8:00 a.m. on January 1st to 8:00 a.m. on January 2nd, the station will probably use January 2nd as the
reporting time. However, two thirds of the 24-h time are within January 1st, indicating that the accumulated
precipitation could mostly occur on January 1st. $T_{max}$ could also occur during the daytime on January 1st, but it is hard
to determine on which day $T_{min}$ occurs, which makes it challenging to adjust precipitation, $T_{min}$ and $T_{max}$ at the same
time. The difference between universal and local time makes this problem more complicated. Thus, the reporting time
of stations is not corrected here due to aforementioned difficulties.
**5.2 Homogenization**
Inhomogeneities in station observations are defined as variations that are not caused by weather and climate factors.
Long-term station records are often subjected to inhomogeneities due to factors like station re-location, observation
time change, instrument change, and surrounding environment change (Venema et al., 2012). Many methods have
been developed to identify breakpoints and homogenize station series in annual, monthly or even daily scales (e.g.,
Ma et al., 2008; Vincent et al., 2002, 2012). Different methods could generate different estimates of inhomogeneities
as shown by many comparison studies (e.g., Beaulieu et al., 2008; Reeves et al., 2007; Venema et al., 2012). The four
station databases (Sect. 2.1) used in this study provide original station records without homogenization. The SCDNA
would inherit the potential inhomogeneities contained in these databases, and the infilling/reconstruction may also
lead to discontinuities. The homogenization of the SCDNA is challenging considering that (1) the dataset covers a
broad range of climate, topography, and countries, (2) the number of stations is large and differences between station
periods (ranging from 8 to 40 years) are substantial, and (3) whether existing methods are suitable for homogenization
of infilling/reconstruction estimates needs exploration. Therefore, homogenization is not carried out in this study,
which, however, is an important direction of future studies.
**5.3 Potential improvement directions**
Several steps could be taken to improve the SCDNA. First, the optimal strategy could be different for each station as
shown by the results in this study. Therefore, the quality of SCDNA may be further improved by using more
infilling/reconstruction methods, which would yield diminishing returns at some point. For example, the long short-
term memory (LSTM) could be suitable to impute missing station observations. Optimizing the configuration of
various strategies will be necessary to balance computation efficiency and estimation accuracy, particularly when the
number of stations is large. Second, some stations suffer from undercatch, which depends on gauge type, precipitation
phase, environmental conditions, etc. The bias caused by undercatch can be substantial for stations located in high
latitudes and in the mountains (Yang et al., 2005; Scaff et al., 2015). Third, the SCDNA does not distinguish between
rainfall and snowfall. Considering that a large part of North America has frequent snowfall in winter, precipitation
phase classification will be useful for hydrometeorological studies. Auxiliary data from reanalysis and satellite
products could be used to partition precipitation into rain and snow. Finally, although the SCDNA agrees well with
station observations, long-term trends are difficult to reconstruct when actual observations are unavailable, meaning
the SCDNA may not be suitable for climate trend analysis in the reconstruction period. Some gridded datasets use





only stations with long-term records (e.g., (Wood, 2008; Werner et al., 2019) to achieve temporally consistent
estimates, whereas such stations are very few. Reasonable trend estimation is challenging but meaningful for SCD.
**6 Data availability**
The SCDNA dataset is available at https://doi.org/10.5281/zenodo.3735534 (Tang et al., 2020) in netCDF format. The
basic variables are station identification, latitude, longitude, elevation, date, and TLR derived in Sect. 3.2. Stations
that undergo location merging (Sect. 3.1.1) are identified and all relevant stations are included in the data file. For
precipitation, $T_{min}$, and $T_{max}$, the variables in the netCDF4 file include original station observations, quality flags
provided by original station databases, quality flags provided by this study, estimates from 16 strategies, uncorrected
SCDNA estimates, corrected SCDNA estimates, the final SCDNA with estimates replaced by observations, data
source flags indicating the source of each record in SCDNA (observations or 16 strategies), and accuracy metrics
(KGE and its three components) for all estimates (16 strategies and SCDNA).
Scripts used to produce the SCDNA are available at https://github.com/tgq14/GapFill. The dataset will be regularly
updated to cover latest periods.
**7 Conclusions**
This study developed a daily SCD of precipitation, $T_{min}$, and $T_{max}$ for 27280 stations from 1979 to 2018 over North
America (SCDNA). The original station data are compiled from multiple sources and undergo strict quality control.
Many stations have nonnegligible fractions of missing values in observation and reconstruction periods. For each
station, the infilling and reconstruction are implemented using 16 strategies (quantile mapping, statistical interpolation,
and machine learning) based on information from neighboring stations and concurrent reanalysis estimates (ERA5,
JRA-55, and MERRA-2). The final SCDNA combines estimates from the 16 strategies and is corrected using station
observations. The spatial correlation is preserved and might be slightly inflated. The SCDNA estimates reproduce the
variance of original station observations very well, particularly for temperature. The median KGE of the final
precipitation, $T_{min}$, and $T_{max}$ for all stations is 0.90, 0.98, and 0.99, respectively. The comparison with four benchmark
gridded products shows that the SCDNA has much better agreement with station observations. The SCDNA will be
useful for a variety of hydrometeorological studies in North America.

**Author contributions:** GT and MC designed the study. GT performed the analyses and wrote the paper. All authors
contributed to data analysis, discussions about the methods and results, and paper improvement.
**Competing interests:** The authors declare that they have no conflict of interest.





**Acknowledgements:** The study is funded by the Global Water Futures (GWF) program in Canada. The authors
appreciate the extensive efforts from the developers of the ground and reanalysis datasets to make their products
available. The authors also thank Zenodo (https://zenodo.org/) for publishing our dataset as open access to users.
**Appendix A**
Table A1. Acronyms used in this paper

| Acronym | Full name |
|---|---|
| ANN | Artificial neural network |
| APHRODITE | Asian Precipitation-Highly-Resolved Observational Data Integration Towards Evaluation |
| CC | Correlation coefficient |
| CDF | Cumulative distribution function |
| CONUS | Contiguous United States |
| DEM | Digital elevation model |
| DOY | Day of year |
| ECCC | Environment and Climate Change Canada |
| ERA5 | the fifth generation of ECMWF atmospheric reanalyses of the global climate |
| fD | Fraction of days without precipitation |
| GHCN-D | Global Historical Climate Network Daily |
| GSOD | Global Surface Summary of the Day |
| IDW | Inverse distance weighting |
| INT | Interpolation |
| JRA-55 | Japanese 55-year Reanalysis |
| KGE | Kling-Gupta Efficiency |
| LSTM | Long short-term memory |
| MAL | Machine learning |
| MLAD | Multiple regression based on the least absolute deviation criteria |
| MERIT DEM | Multi-Error-Removed Improved-Terrain digital elevation model |
| MERRA-2 | Modern-Era Retrospective analysis for Research and Applications, Version 2 |
| MRG | Multi-strategy merging |
| MSWEP | Multi-Source Weighted-Ensemble Precipitation |
| NR | Revised normal ratio |
| PCC | Pearson CC |
| QM | Quantile mapping |
| QMN | QM using neighboring stations |
| QMR | Quantile mapping with concurrent reanalysis estimates |
| RF | Random forest |
| SCC | Spearman CC |



| SCDs | Serially complete datasets |
|---|---|
| TLR | Temperature lapse rate |
| $T_{max}$ | Maximum temperature |
| $T_{mean}$ | Mean temperature |
| $T_{min}$ | Minimum temperature |
| $T_{range}$ | Daily temperature range |
| U.S. | United States |
| UTC | Universal Time Coordinated |


**Appendix B**

Five types of checks (Durre et al., 2010) are adopted for the quality control of temperature.
1.  Integrity checks. The first type of integrity check is *a duplication check* to identify duplicated records for time
series in different time periods. The second type of integrity check includes *the streak check* to identify
consecutive identical values and *the frequent-value check* to identify close but not necessarily consecutive
identical values. The *world record exceedance check* sets lower (-89.4°C) and upper (57.7°C) bounds of
temperature.

2.  Outlier checks, including *the gap check* that examines the frequency distributions for all calendar months, and
the *climatological outlier check* that is based on the traditional z-score (e.g., Hubbard and You, 2005).

3.  Internal and temporal consistency checks, including *the iterative temperature consistency check*, to ensure some
inherent relationships are abided (e.g., $T_{min}$ cannot be larger than $T_{max}$); *the spike/dip check*, identifies
temperatures which deviate from previous and following days by at least 25°C; and *the lagged temperature range*
*check*, which identifies abnormally large differences between $T_{min}$ and $T_{max}$ during a 3-day time window.

4.  Spatial consistency checks, including *the regression check* and *the spatial corroboration check. The regression*
*check* builds regression relationships between temperature at the target location and selected nearby stations to
determine whether temperature at the target station should be flagged according to regression residuals and
standardized residuals. *The spatial corroboration check* flags temperature at the target station if the value
deviates far from the temperature at neighboring stations.

5.  Extreme megaconsistency checks to ensure that certain relationships hold for the entire records of stations. For
example, $T_{max}$ cannot be higher than the lowest $T_{min}$ for the calendar month, and vice versa.

For precipitation, quality control strategies are from three studies. The first part is similar with temperature, but does
not include the third type of checks (internal and temporal consistency checks). The second part is from Hamada et al.
(2011).



1.   Repetition checks. The non-zero check identifies constant daily values (> 10 mm d$^{-1}$) that occur for more than
four days. The zero check compares the annual zero-precipitation frequency with its climatological value to spot
unusual frequencies of zero.

2.   Duplicated monthly or sub-monthly record check. The temporal CC and the number of days with equal
precipitation are used to identify whether two different months have the same records caused by human errors.

3.   Z-score-based outlier check. Daily precipitation is flagged if its difference with the mean value from precipitation
within a 15-day window of all years is larger than nine standard deviations. This step is repeated until no outlier
is identified.

4.   Spatiotemporally isolated value check. Extremely large precipitation is identified in both space and time based
on the percentiles of precipitation differences between the target station and neighboring stations within a radius
of 400 km.

The third part is from Beck et al. (2019).
1.   Empirical criterion based on the fraction of days without precipitation ($f$D). This was designed to identify the long
series of erroneous zero precipitation contained in GSOD station records. However, we found that this criterion
misidentifies some acceptable records in GHCN-D. Therefore, the $f$D-based check is only implemented for GSOD.

2.   Discarding stations with fewer than 15 unique values or more than 99.5% dry records (<0.5 mm d$^{-1}$).

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
