# Peer review of "SCDNA: a serially complete precipitation and temperature dataset for North America from 1979 to 2018"

_Earth System Science Data, 2020_

## Referee Comment (RC1) · Anonymous Referee #1 · 6 Jul 2020

**General comment**

The manuscript presents and advertises a very interesting dataset of temperature and precipitation observation collected over several years in North America. The work is certainly well suited for the readership of ESSD and it is overall very important for the meteorological and climatological community. Furthermore, creation of quality-controlled databases is an important contribution to the scientific community in the age of data science. I have a few points to consider before publication, which i recommend, listed below.

1. **Measurement instruments**: from my background, i am much closer to the instruments themselves (and their peculiarities and issues), as hardware tools. What i missed here was a description of the stations and their instruments. Questions like: which are the instruments deployed in the stations? How is precipitation measured (tipping buckets? buckets? Weighing gauges? Note for example that some instruments may have biases when measuring snowfall while others may not) ? How is it temperature measured? How is this different from station to station in your database?

2. **Codes**: have you considered adding a little reader with a few capabilities, as additional tool for the interested users?

**Minor/Details**

1. P2: as trivial as it can be, it is worth to define the term "station".

2. P3, L96: Why exactly the variables of $T_{min}$, $T_{max}$, and precipitation have been chosen? Is it a matter of (lack of) availability of other measurements? (humidity, wind, etc). I just suggest to clarify.

3. Is precipitation the daily amount? I probably missed this information.

---

## Referee Comment (RC2) · Anonymous Referee #2 · 15 Jul 2020

This study develops a very useful dataset (SCDNA) of serially complete precipitation and temperature in North America. The dataset will benefit researchers in various fields with the long-term and gap-filled station data collected from multiple sources. The sophisticated framework for imputing missing values is well designed, which can be potentially applied in other regions of the world for the production of regional or even global serially complete datasets. From my perspective, the paper can be published on ESSD after the minor revisions, and I also have a few comments as below. 1. The differences between SCDNA and MSWEP show distinct differences along the boundaries of CONUS and Canada. Can you provide more detailed explanation about how observation time inconsistency causes this problem? 2. The paper said "Outputs from three

reanalysis products (ERA5, JRA-55, and MERRA-2) provided auxiliary information to estimate station records and were also used as an assessment benchmark. ". Can you give more explanation why you selected reanalysis products for benchmark? 3. The period from 1979 to 2018 is total 40 years. Numbers of stations with only at least 8-year records are shown in Table 1. Why only 8-year period records are showed? Are only stations with at least 8-year precipitation or Tmin and Tmax records between 1979 to 2018 utilized to evaluate the performance? Is there some difference between 8-year records and total records for evaluation? 4. Precipitation and minimum/maximum temperature are very widely used in hydrometeorological studies. I think probably this is why the three variables are chosen. Considering meteorological stations can usually measure more variables which also suffer from missing values, expanding this work to other variables would be very interesting for future studies. I suggest that the authors add some discussion about the applicability of your method to other variables.

---

## Author Comment (AC1) · 31 Jul 2020

**Response to comments**

The authors thank the reviewers for their constructive comments, which provide the basis to improve the quality of the manuscript and dataset. We address all points in detail and reply to all comments here below. We also updated SCDNA from V1 to V1.1 on Zenodo based on the reviewer's comments. The modifications include adding station source flag, adding original files for location merged stations, and adding a quality control procedure based on the final SCDNA. SCDNA estimates are generally consistent between the two versions, with the total number of stations reduced from 27280 to 27276.

**Reviewer 1**

**General comment**

The manuscript presents and advertises a very interesting dataset of temperature and precipitation observation collected over several years in North America. The work is certainly well suited for the readership of ESSD and it is overall very important for the meteorological and climatological community. Furthermore, creation of quality controlled databases is an important contribution to the scientific community in the age of data science. I have a few points to consider before publication, which I recommend, listed below.

1. Measurement instruments: from my background, I am much closer to the instruments themselves (and their peculiarities and issues), as hardware tools. What I missed here was a description of the stations and their instruments. Questions like: which are the instruments deployed in the stations? How is precipitation measured (tipping buckets? buckets? Weighing gauges? Note for example that some instruments may have biases when measuring snowfall while others may not)? How is it temperature measured? How is this different from station to station in your database?

Response: We have added the descriptions of measurement instruments in both the manuscript and dataset documentation. Since a complete introduction to the specifications and the evolution of measurement instruments in North America is not trivial, we only provide a general introduction here, and guide readers to the official sources for more comprehensive knowledge (such as design purpose, instrument structure, accuracy for rain/snow, inter-instrument comparison) in the manuscript and dataset page. As station hardware varies among countries, we successively introduce the overall situations in Canada, U.S., and Mexico as below.

For Canada, the Type-B rain gauge is used since 1970s for most stations by Environment Canada (Devine and Mekis, 2008; Wang et al., 2017). Tipping bucket and weighing gauges are also used

in some stations (Metcalfe et al., 1997). For snowfall measurement, Nipher-shielded snow gauges were introduced at nearly 300 synoptic stations in the early 1960s, while most snow observation stations still rely on ruler measurements (https://www.canada.ca/en/environment-climate-change/services/sky-watchers/weather-instruments-tour.html). For temperature, weather observers use as many as 4 different thermometers mounted inside the Stevenson screen. Maximum and minimum thermometers use mercury and alcohol, respectively (https://www.canada.ca/en/environment-climate-change/services/sky-watchers/weather-instruments-tour/thermometers-thermistors.html). However, detailed metadata for an individual station is hard to obtain (e.g., see the detailed analysis of Whitfield (2014) for the station 3053600 in Kananaskis, Alberta).

For the U.S.A., station data are provided by many agencies/programs. The sources are denoted in SCDNA using the source flags provided in the GHCN-D dataset. For stations from the Cooperative Observer Program (COOP), the instruments are summarized in https://www.weather.gov/ilx/coop-equipment. The Standard Rain Gage (SRG) is used, and the method for measuring rainfall and snowfall is summarized in https://www.weather.gov/iwx/coop_8inch. For stations from Community Collaborative Rain, Hail,and Snow (CoCoRaHS), a 4-inch diameter rain gauge is used (https://www.cocorahs.org/Content.aspx?page=rain). For the U.S. Automated Surface Observing System (ASOS), heating Heated Tipping Bucket (HTB) and hygrothermometer are used for most stations, and there is a transition from HTB to All Weather Precipitation Accumulation Gauge (AWPAG) since 2004 (https://www.weather.gov/asos/ASOSImplementation, file:///Users/localuser/Downloads/ASOS_guide_1998.pdf). For NCEI Reference Network Database, a combination of weighing gauge, precipitation detector, and tipping bucket gauge are used, and air temperature is measured using three platinum resistance thermometers housed in fan aspirated solar radiation shields (https://www.ncdc.noaa.gov/crn/instruments.html). For SNOTEL, storage-type gage or tipping bucket is used, and temperature is measured using shielded thermistor (https://www.wcc.nrcs.usda.gov/snotel/snotel_sensors.html, https://www.wcc.nrcs.usda.gov/about/mon_automate.html). For Remote Automatic Weather Station (RAWS), THS-3 temperature and humidity sensor and RG-T tipping bucket rain gauge are used (https://www.fs.fed.us/eacc/library/docs/RAWS_WIMS_Guide.pdf, https://ftsinc.com/fixed-remote-automated-weather-station). For High Plains Regional Climate Center real-time data, tipping bucket or rain gauge is used (https://hprcc.unl.edu/awdn/index.php).

For Mexico, the automatic weather station, which is a set of electrical and mechanical devices that perform measurements of meteorological variables automatically (WMO Reference 182) are used by Servicio Meteorológico Nacional. (https://smn.conagua.gob.mx/es/observando-el-tiempo/estaciones-meteorologicas-automaticas-ema-s).

A useful database, the Historical Observing Metadata Repository (HOMR), is maintained by NOAA NCEI (https://www.ncdc.noaa.gov/data-access/land-based-station-data/station-metadata).

Users can find detailed information of a station using station ID provided by different station sources, including SCDNA. For example, COOP station USC00244302 measures precipitation using SRG from 2000 to 2018-10-4 and SRG-STN since 2018-10-4. However, instrument information could be missing for many stations outside U.S.

We added a paragraph in Section 2.1: "Many types of precipitation and temperature measurement instruments are used at stations from different sources. For example, the Type-B rain gauge is used by Environment Canada since 1970s for most weather stations (Devine and Mekis, 2008; Wang et al., 2017), while tipping bucket and weighing rain gauges are also used in some stations (Metcalfe et al., 1997). Nipher-shielded snow gauges have been used by some synoptic stations, while ruler measurements are still used by more stations (Mekis and Brown, 2010). Station data in U.S. are from many organizations or programs with different instrument configurations. For instance, the standard rain gauge is used by the Cooperative Observer Program while Snow Telemetry uses storage-type gauges or tipping buckets. A better understanding of instrument specifications and historical changes is important for climate studies (Pielke Sr et al., 2007; Whitfield, 2014; Ma et al., 2019). A detailed summary of station instruments is provided in the documentation of the dataset (https://doi.org/10.5281/zenodo.3953310)."

Reference:

Devine, K. A., & Mekis, E. (2008). Field accuracy of Canadian rain measurements. Atmosphere-ocean, 46(2), 213-227.

Mekis, É., & Brown, R. (2010). Derivation of an adjustment factor map for the estimation of the water equivalent of snowfall from ruler measurements in Canada. Atmosphere-ocean, 48(4), 284-293.

Metcalfe, J. R., B. Routledge, and K. Devine. 1997. Rainfall measurement in Canada: Changing observational methods and archive adjustment procedures. Journal of Climate 10: 92-101.

Pielke Sr, R., Nielsen-Gammon, J., Davey, C., Angel, J., Bliss, O., Doesken, N., ... & Hale, R. (2007). Documentation of uncertainties and biases associated with surface temperature measurement sites for climate change assessment. Bulletin of the American Meteorological Society, 88(6), 913-928.

Whitfield, P. H. 2014. Climate station analysis and fitness for purpose assessment of 3053600 Kananaskis, Alberta. Atmosphere-Ocean 52(5): 363-383.

Wang, X. L., Xu, H., Qian, B., Feng, Y., & Mekis, E. (2017). Adjusted daily rainfall and snowfall data for Canada. Atmosphere-Ocean, 55(3), 155-168.

2. Codes: have you considered adding a little reader with a few capabilities, as additional tool for the interested users?

Response: We have added more detailed descriptions on GitHub (https://github.com/tgq14/GapFill/blob/master/README.md). The functions and their usage of different modules are introduced in Readme.md. Users can utilize the entire or part of the code package with the help of comments contained in scripts.

Minor/Details

1. P2: as trivial as it can be, it is worth to define the term "station".

Response: We added the definition. The revised sentence in P2 is "Many methods have been developed to estimate missing observations and reconstruct time series of meteorological stations that provide point-scale regular observations of atmospheric conditions".

2. P3, L96: Why exactly the variables of Tmin, Tmax, and precipitation have been chosen? Is it a matter of (lack of) availability of other measurements? (humidity, wind, etc). I just suggest to clarify.

Response: We selected the three variables for two reasons. First, as you have indicated, precipitation, Tmin and Tmax are the most common variables provided by meteorological stations, while other variables such as wind or humidity are less common. Second, most previous studies focus on precipitation and temperature, while other variables attract less attention. Thus, whether our methodology will work for other variables needs further investigation. We added explanation in the first paragraph in P3: "The three variables are selected because (1) most stations measure precipitation and temperature, while other variables, such as humidity and wind speed are measured at fewer stations, and (2) precipitation and temperature data are fundamental inputs for hydrological modeling."

We also added discussion on involving other variables in future work in Section 5.3.

3. Is precipitation the daily amount? I probably missed this information.

Response: Yes, it is. We added explanation in the first paragraph in Section 2.1: "In this dataset, precipitation is the daily amount."

**Reviewer 2**

This study develops a very useful dataset (SCDNA) of serially complete precipitation and temperature in North America. The dataset will benefit researchers in various fields with the long-term and gap-filled station data collected from multiple sources. The sophisticated framework for imputing missing values is well designed, which can be potentially applied in other regions of the world for the production of regional or even global serially complete datasets. From my perspective, the paper can be published on ESSD after the minor revisions, and I also have a few comments as below.

1. The differences between SCDNA and MSWEP show distinct differences along the boundaries of CONUS and Canada. Can you provide more detailed explanation about how observation time inconsistency causes this problem?

Response: MSWEP merges data from satellite products, reanalysis models and ground observations. Station data in different regions could have different observation time. To match station and reanalysis/satellite data, MSWEP calculates daily grid- and gauge-based time series, with the grid-based time series shifted by offsets of −36, −33, −30, …, +30, +33, and +36 h. Then, the temporal offset with the highest correlation is used to calculate 24-h accumulation of daily precipitation (Beck et al., 2019). Therefore, the final MSWEP estimates do not necessarily correspond to the raw observation of stations. For CONUS and Canada, the temporal offset is different and thus the mismatch between MSWEP and original station data is different.

We added an explanation in the third paragraph in Section 4.4: "Fig. 15 shows notable differences between MSWEP and SCDNA at the Canada-USA border and the USA-Mexico border. This is because MSWEP infers gauge reporting time by searching for the highest correlation between gauge data and the temporally shifted reanalysis/satellite estimates (Beck et al., 2019). The estimated temporal shift could vary with countries, which results in distinct differences of station-based evaluation results along national boundaries."

Reference:

Beck, H. E., Wood, E. F., Pan, M., Fisher, C. K., Miralles, D. G., Van Dijk, A. I., ... & Adler, R. F. (2019). MSWEP V2 global 3-hourly 0.1 precipitation: methodology and quantitative assessment. Bulletin of the American Meteorological Society, 100(3), 473-500.

2. The paper said "Outputs from three reanalysis products (ERA5, JRA-55, and MERRA-2) provided auxiliary information to estimate station records and were also used as an assessment benchmark. ". Can you give more explanation why you selected reanalysis products for benchmark?

Response: We choose the three products because (1) they are produced by representative reanalysis models from organizations in U.S., Europe, and Japan, and (2) they or their predecessors (ERA-Interim, JRA-25, and MERRA) have are been widely used by previous studies (e.g., Sun et al., 2018). The three reanalysis products are used as benchmark because they are widely used as the source of long-term precipitation and temperature data and have been applied to support infilling and reconstruction in this study.

We added an explanation in Section 2.2: "The three products are chosen because they are representative products from different international organizations and they or their predecessor (ERA-Interim, JRA-25, and MERRA) have are been widely used by researchers.".

Reference:

Sun, Q., Miao, C., Duan, Q., Ashouri, H., Sorooshian, S., & Hsu, K. L. (2018). A review of global precipitation data sets: Data sources, estimation, and intercomparisons. Reviews of Geophysics, 56(1), 79-107.

3. The period from 1979 to 2018 is total 40 years. Numbers of stations with only at least 8-year records are shown in Table 1. Why only 8-year period records are showed? Are only stations with at least 8-year precipitation or Tmin and Tmax records between 1979 to 2018 utilized to evaluate the performance? Is there some difference between 8-year records and total records for evaluation?

Response: For the first question, we only show 8-year records because according to our sensitivity analysis, eight years are enough to ensure gap filling is generally reliable (Figure S1). Using a higher period threshold can improve the quality of the final dataset but will reduce the number of stations.

For the second question, yes, only stations with at least eight-year records are used for evaluation to be consistent with inputs.

For the third question, our evaluation is based on 30% samples of each station. For example, if a station has 8-year/40-year observations, the validation samples are about 2.4-year/12-year. Therefore, the evaluation period length could be different for different stations. According to our results (Figures 6 and 12), the spatial distributions of accuracy metrics and contribution ratios are smooth, indicating that the difference between 8-year records and total records for evaluation is not evident. We added explanation in Step-5 in Section 3.3.3: "Although the evaluation samples are different among stations, the results are reliable and stable as shown in the results section."

4. Precipitation and minimum/maximum temperature are very widely used in hydrometeorological studies. I think probably this is why the three variables are chosen. Considering meteorological stations can usually measure more variables which also suffer from missing values, expanding this

work to other variables would be very interesting for future studies. I suggest that the authors add some discussion about the applicability of your method to other variables.

Response: Thank you for this suggestion. Expanding this work to other variables will be an interesting study. We added discussion in Section 5.3: "Furthermore, other variables such as wind and humidity observed by stations also suffer from the same problems faced by precipitation and temperature. Future studies should explore whether the current methodology is applicable to other variables. A SCD covering more variables would be useful for research in various fields."